



# The impact of atmospheric dynamics on vertical cloud overlap over the Tibetan Plateau

Jiming Li[1,*], Qiaoyi Lv[2], Bida Jian[1], Min Zhang[1],

Chuanfeng Zhao[3], Qiang Fu[1,4], and Kazuaki Kawamoto[5], Hua Zhang[6]

[1]Key Laboratory for Semi-Arid Climate Change of the Ministry of Education, College of Atmospheric Sciences, Lanzhou University, Lanzhou, China

[2]Laboratory of Straits Meteorology, Xiamen Meteorological Bureau, Xiamen, China

[3]State Key Laboratory of Earth Surface Processes and Resource Ecology, and College of Global Change and Earth System Science, Beijing Normal University, Beijing, China

[4]Department of Atmospheric Sciences, University of Washington, Seattle, USA

[5]Graduate School of Fisheries and Environmental Sciences, Nagasaki University, Nagasaki, Japan

[6]Laboratory for Climate Studies, National Climate Center, China Meteorological Administration, Beijing, China

Corresponding author: Jiming Li, Key Laboratory for Semi-Arid Climate Change of the Ministry of Education, College of Atmospheric Sciences, Lanzhou University, Lanzhou, Gansu 730000, China. (lijiming@lzu.edu.cn)

**Abstract**

The accurate representation of cloud vertical overlap in atmospheric models is particularly significant for predicting the total cloud cover and for the calculations related to the radiative budget in these models. However, it has received too little attention due to the limited observation, especially over the Tibetan Plateau (TP). In this study, 4 years (2007–2010) of data from the CloudSat cloud product and collocated ERA-Interim reanalysis product were analyzed to examine the seasonal and zonal variations of cloud overlap properties over the TP region, and evaluate the effect of atmospheric dynamics on cloud overlap.

Unique characteristics of cloud overlap over the TP have been found. The statistical



results show that the random overlap assumption slightly underestimates the total cloud coverage for discontinuous cloud layers over the TP, whereas the overlap parameter $\alpha$ for

continuous cloud sharply decrease from maximum to random overlap with an increase of layer distance, eventually trending towards a minimal overlap (e.g., negative $\alpha$ values ) as the cloud layer separation distance exceeds 1.5 km. Compared with the global averaged cloud overlap characteristics, the proportion of minimal overlap over the TP is significant high (about 41%). It may be associated with the unique topographical forcing

and thermos-dynamical environment of the TP. As a result, we propose a valid scheme for quantifying the degree of cloud overlap over the TP through a linear combination of the maximum and minimum overlap, and further parameterize decorrelation length scale $L$ as a function of wind shear and atmospheric stability. Compared with other parameterizations, the new scheme reduces the bias between predicted and observed

cloud covers. These results thus indicate that effects of wind shear and atmospheric stability on cloud overlap should both be taken into account in the parameterization of overlap parameter $\alpha$ to improve the simulation of total cloud cover in models.

## 1. Introduction

Clouds can cause considerable changes in the Earth's radiation budget, the global hydrological cycle and large-scale atmospheric circulations via changes in their various macrophysical (e.g., cloud cover, height, and thickness) and microphysical properties (e.g., cloud phase and droplet and crystal size) (Rossow and Lacis, 1990; Hartmann et al., 1992; Stephens, 2005; Kawamoto and Suzuki, 2012; Yan et al., 2012; Wang et al., 2010).

However, our incomplete understanding of their underlying physical processes makes the representation of clouds in climate models still unreliable, which keeps clouds as the largest uncertainty when estimating and interpreting changes in the Earth's energy budget (Boucher et al., 2013).

The Tibetan Plateau (TP), which is also known as the "roof of the world" or the

"world water tower", plays a significant role in determining global atmospheric circulations, in addition to its strong influence over Asia via its thermal and dynamic forcings (Yanai et al., 1992; Ye and Wu, 1998; Duan and Wu, 2005; Xu et al., 2008; Wu et al., 2015). Specifically, the TP has experienced distinct climate changes over the past




three decades (Kang et al., 2010) that have changed its atmospheric and hydrological

cycles (Yang et al., 2014). Some studies have linked these climate changes to variations in the cloud cover over the TP region (e.g., Chen and Liu, 2005; Duan and Wu, 2006; Li et al., 2006; Yang et al., 2012; You et al., 2014; Wu et al., 2014; Duan and Xiao, 2015). However, our understanding about the role of cloud cover on the radiation balance and water cycle over the TP region remains poor because of the limited availability of

regional cloud observations and our incomplete knowledge of the cloud physical processes for use in weather forecasting and climate models. One of the remaining challenges involves how to reasonably represent the characteristics of the vertical overlapping of cloud layers in these models. The different cloud overlap states (e.g., whether it is at a maximum, minimum or random state) used in the models may result in

distinctly different total cloud coverage. For example, we assume that the clouds in each of two model layers produce 50% cloud coverage. As such, the maximum overlap will result in a total cloud coverage of 50%, and a minimum overlap will result in an overcast state (a complete cloud coverage, i.e., 100%). The differences between the predicted total cloud coverages will also significantly affect the calculated radiative budgets and

heating/cooling rate profiles in the model simulations (Morcrette and Fouquart, 1986; Barker et al., 1999; Chen et al., 2000; Pincus et al., 2005; Zhang et al., 2013a; 2013b; 2016; Jing et al., 2016). Previously published general circulation model (GCM) simulation results have indicated that the bias in the global mean radiation fluxes at the top of the atmosphere and at the surface, which is caused by the different overlap

conditions only, can reach 20-40 W m$^{-2}$ (Morcrette and Jakob, 2000; Jing et al., 2009; Zhang and Jing, 2010).

Ground-based radar observations can be used to improve the cloud overlap assumptions in the models because radar signals produce accurate cloud mask profiles (e.g., Hogan and Illingworth, 2000; Mace and Benson-Troth, 2002; Willén et al., 2005;

Naud et al., 2008; Oreopoulos and Norris, 2011). However, these observations are only one-dimensional, and the radar sites are very sparsely distributed, especially in the TP region. Passive sensors and traditional surface weather reports fail to detect vertical cloud structures, providing limited information about the cloud overlap (Chang and Li, 2005a, b; Huang, 2006; Huang et al., 2005, 2006a). Therefore, an active space-borne radar



measurement is the best choice for identifying the vertical overlap properties of clouds and thereby, better quantifying the radiative budget at the surface and its related climate change impacts (e.g., glacier degradation and frozen soil ablation) over the TP region.

Fortunately, the millimeter-wavelength cloud profiling radar (CPR) launched on CloudSat (Stephens et al., 2002) and the cloud-aerosol lidar with orthogonal polarization (CALIOP) (Winker et al., 2007) launched on CALIPSO (Cloud-Aerosol Lidar and Infrared Pathfinder Satellite Observation) provide an unprecedented opportunity to investigate vertical cloud overlaps on a global scale (Barker et al., 2008; Kato et al., 2010; Li et al., 2011; 2015; Tompkins and Di Giuseppe, 2015; Di Giuseppe and Tompkins, 2015). Based on two months of cloud mask profile information from the CloudSat and CALIPSO satellites, Barker (2008) quantified the properties of cloud overlap on a global scale and identified a latitudinal dependence. Recently, Di Giuseppe and Tompkins (2015) further evaluated the impact of wind shear on the global-scale cloud overlap and identified an empirical relationship between the cloud overlap and wind shear for use in models by using 6 months of CloudSat-CALIPSO data. However, the related question of the cloud overlapping over the TP region has received too little attention, as has the question of how the unique thermo-dynamical environment (e.g., the atmospheric stability) of the TP region affects cloud overlap. As such, this study mainly focuses on the impacts of various atmospheric states and large-scale atmospheric dynamics on cloud overlap over the TP region by combining the cloud cover profile information from the 2B-GEOPROF-LIDAR dataset (Mace et al., 2009) and the meteorological parameters from the ERA-Interim reanalysis datasets (Dee et al., 2011).

This paper is organized as follows: a brief introduction to all the datasets and methods used in this study is given in Section. 2. Section 3.1 outlines the seasonal and zonal variations of the cloud overlap parameters over the TP region. Further analyses of the impacts of the variations in the atmospheric state and large-scale atmospheric dynamics on cloud overlap are provided in Sections. 3.2 and 3.3. Finally, the conclusions and discussion are presented in Section. 4.

## 2. Datasets and methodology

In this study, 4 years (2007–2010) of data from CloudSat 2B-GEOPROF-LIDAR, ECMWF-AUX and the daily 6-hour ERA-Interim reanalysis were collected to analyze



the impacts of atmospheric dynamics on the cloud overlap over the TP (27 °N-39 °N;78 °E-103 °E) region as demonstrated in Fig. 1a.

## 2.1 Satellite datasets

Radar signals can penetrate the optically thick layers that attenuate lidar signals significantly, and lidar signals may sense the optically thin hydrometeor layers that are below the detection threshold of radar signals. Thus, by combining the unique complementary capabilities of the CPR on CloudSat and the space-based polarization lidar (CALIOP) on the CALIPSO satellite, the 2B-GEOPROF-LIDAR dataset can produce the most accurate quantitative descriptions of the locations of the hydrometeor layers in the atmosphere on the global scale (Mace and Zhang, 2014). For this study, we only use the "*CloudFraction*" parameter from the aforementioned dataset. This parameter reports the fraction of the lidar volume within the radar resolution volume that contains hydrometeors (Mace et al., 2009; Mace and Zhang, 2014) and describes the cloud mask profile. Several previous studies have identified a cloudy atmospheric layer based on different thresholds of the lidar-identified cloud fraction, including a 99% (Barker, 2008; Di Giuseppe and Tompkins, 2015) or 50% threshold (Haladay and Stephens, 2009; Verlinden et al., 2011). Although a threshold of 99% is used in this study, we also test the sensitivity of cloud overlap to the cloud fraction threshold. In addition, the ECMWF-AUX dataset (Partain, 2004), which is an intermediate dataset that contains the set of ancillary ECMWF state variable data interpolated across each CloudSat CPR bin, are also used to provide the pressure and height information of each vertical bin in the cloud mask profile. The vertical and horizontal resolutions of these products are 240 m and 1.1 km, respectively.

## 2.2 Meteorological reanalysis dataset

The daily 6-hourly dataset from the ERA-Interim reanalysis (Dee et al., 2011), which has a grid resolution of 0.25 °×0.25 °, is used to characterize the atmospheric dynamics over the TP. For each cloud mask profile in the 2B-GEOPROF-LIDAR dataset, the vertical profiles of the zonal wind $u$, meridional wind $v$, relative humidity $rh$, specific humidity $sh$ and atmospheric temperature $T$ most closest to the cloud observations in both space and time are extracted and further interpolated in the vertical direction to match the bin numbers and sizes of the cloud mask profile. Then, we follow the method used by Di





Giuseppe and Tompkins (2015) to project the $u$ and $v$ winds of every vertical bin onto the satellite overpass track, averaging in the along-track direction for all profiles in the selected CloudSat data segment (see section 2.3) to derive the scene-average, along-track

160     horizontal wind $V$ and the corresponding wind shear. Here, we define the wind shear $dV/dz_{i,j}$ between the layers $i$ and $j$, which is used to calculate the cloud overlap parameter as follows:

$$dV/dz_{i,j} = \frac{\max\{V_i; V_j\} - \min\{V_i; V_j\}}{D_{i,j}} \quad , \tag{1}$$

where $V_i$ and $V_j$ are the horizontal winds at layers $i$ and $j$, respectively, and $D_{i,j}$ is the layer

165     separation distance. For the CloudSat overpass track (Fig. 1a), Di Giuseppe and Tompkins (2015) indicated that the cross-track shear of the zonal wind $u$ has little statistical significance.

Similarly with the wind shear, we calculate the vertical gradient of the saturation of the equivalent potential temperature ($\partial \theta_{es}/\partial z_{i,j}$) between the same two layers to quantify

170     the dependence of the cloud overlap on the degree of the conditional instability of the moist convection. Here,

$$\theta_{es} = \theta \exp(\frac{L_v r_s}{C_p T})$$
$$\theta = T(\frac{1000}{p})^{0.286}, L_v = 2.5 \times 10^6 - 2323 \times (T - 273.16) \tag{2}$$
$$r_s = \frac{sh}{rh \times (1 - sh)}$$

where $\theta$ is the potential temperature, $L_v$ is the latent heat of vaporization, $r_s$ is the saturation mixing ratio, $C_p$ is the specific heat capacity at a constant pressure, and $T$ is the

175     atmospheric temperature. The smaller the $\partial \theta_{es}/\partial z_{i,j}$, the more unstable the atmosphere. Additionally, the scene-averaged vertical velocity at 500 hPa is extracted from the ERA-Interim reanalysis to analyze the impact of vertical motion on cloud overlap.

### 2.3 Retrieval of the overlap parameter

Each CloudSat orbit over the TP region is divided into different segments based on

180     the horizontal length of the segment (e.g., 25, 50, 100 and 200 km; hereafter, this length is referred to as the spatial sampling scale or domain size). Fig 1b shows a sample cloud



mask over the TP region derived from the 2B-GEOPROF-lidar dataset. This cloud mask includes eight, four, two and one segments with lengths of 25, 50, 100 and 200 km, respectively. For each segment, the average cloud cover profile is derived first. Then the vertical overlap between any two atmospheric layers in this profile is calculated when the cloud coverages ($C_i$ and $C_j$) of both layers exceed 0. Following the examples of previous studies (Hogan and Illingworth, 2000; Di Giuseppe and Tompkins, 2015), we consider nonadjacent layers to be a continuous cloud pair when all layers between them are classified as cloudy. Otherwise, these layers are classified as a discontinuous cloud pair (Hogan and Illingworth, 2000; Di Giuseppe and Tompkins, 2015). Using various assumptions of overlap, the total cloud coverages of any two cloud layers are given by

$$
\begin{aligned}
C_{i,j}^{ran} &= C_i + C_j - C_i \times C_j, \\
C_{i,j}^{max} &= \max\{C_i, C_j\}, \\
C_{i,j}^{min} &= \min\{1, C_i + C_j\},
\end{aligned}
\tag{3}
$$

Further, we use the definition of Hogan and Illingworth (2000) to quantify the degree of cloud overlap using the "overlap parameter" $\alpha$, which is calculated using the following formula:

$$
C_{i,j}^{obs} = \alpha C_{i,j}^{max} + (1-\alpha) C_{i,j}^{ran}
\tag{4}
$$

The overlap parameter $\alpha$ can thus be written as follows:

$$
\alpha = \frac{C_{i,j}^{obs} - C_{i,j}^{ran}}{C_{i,j}^{max} - C_{i,j}^{ran}}
\tag{5}
$$

Eq. 4 indicates that the parameter $\alpha$ ranges from 0 (random) to 1 (maximum) when the total observed cloud cover falls between the values derived using the maximum and random overlap assumptions. However, $\alpha$ may be negative when the degree of cloud overlap is lower than that predicted by the random overlap assumption. Fig. 2 tests the sensitivity of cloud overlap to the chosen cloud fraction threshold by using discontinuous and continuous cloud pairs with a 50-km spatial sampling scale. The value of $\alpha$ decreases as the cloud fraction threshold increases, especially for discontinuous cloud pairs. However, this sensitivity decreases when the cloud fraction threshold exceeds 50%. Given these results, a cloud fraction threshold of 99% is used in the following sections. Using this cloud fraction threshold (99%), we test the sensitivity of $\alpha$ to the spatial





sampling scale for both discontinuous and continuous cloud pairs (Fig. 3a and Fig. 3c).

Many previous studies have used ground- and space-based radar to verify the validity of the random overlap assumption for the study of vertically discontinuous clouds (Hogan and Illingworth, 2000; Mace at al., 2002; Naud et al., 2008; Di Giuseppe and Tompkins, 2015). Our results show that the degree of cloud overlap of the discontinuous clouds over the TP region is lower than that of random overlap. In Fig. 3a, almost all the $\alpha$ values

are negative at the given cloud fraction threshold (99%) and fall between -0.2 and 0. Thus, the total cloud coverage is still slightly underestimated for discontinuous cloud pairs when using the random overlap assumption. Assuming a cloud layer distance of less than 9 km, $\alpha$ increases as the spatial sampling scale increases (e.g., from 25 km to 200 km). This correlation indicates that a maximum overlap is more common for a larger domain,

which is consistent with the results of previous studies (Hogan and Illingworth, 2000; Oreopoulos and Khairoutdinov, 2003; Oreopoulos and Norris, 2011). The obvious dependence of $\alpha$ on the spatial sampling scale is also observed for continuous cloud pairs when they are separated by a distance of less than 9 km (Fig. 3c). For a continuous cloud pair, we found that the value of $\alpha$ decreased from 0.92 to -0.35 and then slightly

increased with increasing distance. When the cloud layer distances exceeded 1.5 km, the degree of cloud overlap was even lower than that seen for random overlap. Our overlap parameters for the TP are clearly different from the globally averaged values derived by Di Giuseppe and Tompkins (2015) although the same datasets (with only 6 months of data) were used. Previous studies have reported the existence of a negative $\alpha$ when the

cloud layer separation distances are greater than 2km, as derived using ground-based radar at different sites (Naud et al., 2008). This result is most often seen for discontinuous cloud layers. The related results also indicated that $\alpha$ and its corresponding decorrelation length are very sensitive to the spatial sampling scale (Hogan and Illingworth, 2000; Mace at al., 2002; Naud et al., 2008; Oreopoulos and Norris, 2011).

However, it is obvious that the more negative $\alpha$ values seen in our statistical results are related to the degree of cloud overlap over the TP, which is more prone to minimal overlap as the cloud layer separation distances exceed 1.5 km. Tompkins and Di Giuseppe (2015) provided some heuristic discussions of the negative $\alpha$ values and sampling scale dependence and attributed these phenomena to a data truncation error.





Simply, these authors theorized that some overcast or single cloud layers will be removed from the samples when the spatial sampling scale is smaller than the cloud system scale, thus biasing $\alpha$ and its decorrelation length. Given a spatial sampling scale of 50 km, the ratio of the sampling scale to the cloud system scale decreases strongly from the equator to the poles because many of the frontal cloud systems of the middle and high latitudes

are larger than the convective cloud systems over the tropics. Thus, regional atmospheric models should account for the typical cloud system scales when they are applied at a fixed horizontal resolution. Based on these arguments, Tompkins and Di Giuseppe (2015) suggested a simple filter for the continuous cloud pairs, so that only measurements with cloud coverages smaller than a given threshold of 50% are retained to reduce the

inversion errors of $\alpha$ and its corresponding decorrelation length caused by data truncation.

Fig. 3b depicts the probability distribution functions (PDFs) of the horizontal scales of the along-track cloud systems at different heights over the TP region. Here, the horizontal scale of a cloud system at a given height along the CALIPSO/CloudSat track is

determined by calculating the number of continuous cloud profiles ($N$) when each profile employs a cloud fraction threshold of 99% at a given height. Using a 1.1 km along-track resolution for the CPR measurements, the along-track scale ($S$ in km) of a cloud system is $S=N\times1.1$ (Zhang et al., 2014; Li et al., 2015). Thus, the probability of a small-scale cloud system decreases with increasing height. On average, the horizontal scale of a

cloud system at a height of 15 km (mean value: 66.4 km) is almost ten times greater than that at a height of 2 km (mean value: 6.7 km). From the statistical results shown in Fig. 3b, we can infer that a spatial sampling scale of 50 km will result in larger retrieval errors of $\alpha$ at higher atmospheric heights above the lower troposphere. Thus, to reduce the uncertainty caused by data truncation, we follow the suggestion of Tompkins and Di

Giuseppe (2015) and apply a data filter, retaining only the cloud samples with cloud coverages of less than 50%. Given the cloud fraction threshold (99%), spatial sampling scale (50 km) and upper limit of cloud cover (50%), the number of available samples is at least one million, thus ensuring statistical significance. Fig. 3d shows the number of samples and the percentages at different cloud layer distances for both discontinuous and

continuous clouds. These results clearly show that the proportion of cloud samples with





smaller layer distances (e.g., 4 km) accounts for 95% of all samples of continuous clouds. Notably, cloud fraction and cloud cover are different variables in our study. "*Cloudfraction*" reports the fraction of lidar volumes in the radar resolution volume that contains hydrometeors and is used to identify a cloudy based on the chosen threshold,

which is 99% in this paper. However, the segment-averaged cloud coverages at certain heights represent the percentage of clouds in a given spatial sampling scale. Given the 1.1 km along-track resolution of the CPR measurements and a spatial sampling scale of 50 km, the segment-averaged cloud coverage implies that the cloud cover is resolvable to approximately 2% (Di Giuseppe and Tompkins, 2015). In addition, to considering the

impact of cloud overlap on the total cloud cover and solar radiation and to minimize the uncertainty of the statistical results from the surface contamination of the CPR, only datasets from 1 km above the TP surface during the daytime were used to perform the analyses in this investigation.

**3. Results**

**3.1 Seasonal and zonal variations of the overlap parameter for continuous clouds**

Figure 4a, 4b and 4c show the seasonal variations for the continuous cloud-pair related pentad-averaged $\alpha$, the degree of conditional instability of the moisture convection $\partial\theta_{es}/\partial z$ and the wind shear $dV/dz$ over the TP, respectively. The maximum cloud layer distance gradually increases from January (approximately 5 km) to August

(approximately 8 km) and then decreases. When the average cloud layer distance is less than 4 km during the summer, the value of $\alpha$ in this season is often larger than the value of $\alpha$ during the other seasons. For example, for a 2-km cloud distance, $\alpha$ reaches its maximum and minimum values in August (0.05) and March (-0.17) (see Fig. 4d). As this distance increases (e.g., to 3 km), $\alpha$ is generally lower but has the similar seasonal

variations to those seen at the 2-km distance. The corresponding $\partial\theta_{es}/\partial z$ and $dV/dz$ both exhibit obvious seasonal variations as well. For layer distances below 4 km, the atmospheric stability and wind shear gradually decrease from January to August and then steadily increase (see Figs. 4b, 4c, 4e and 4f). These trends may be associated with the unique topographical forcing and thermo-dynamical environment of the TP. The TP is

usually considered to be an atmospheric heat source during the summer (Wu et al., 2015). Additionally, at the start of the Asian summer monsoon, abundant water vapor converges,





which intensifies the atmospheric instability of the TP region when combined with the enhanced surface heating (Taniguchi and Koike, 2008). This process further promotes the development of convective clouds. Although similar seasonal variations of $\partial\theta_{es}/\partial z$

occur at different layer distances, Fig. 4e clearly shows a trend of increasing atmospheric instability between the two cloud layers with decreasing layer distance. Using ground-based radars, previous studies have shown that the cloud overlap parameters at different locations exhibit obvious seasonal variations. These studies have attributed the seasonal variations to the effects of meteorological conditions (Hogan and Illingworth,

2000; Mace and Benson-Troth, 2002). Naud et al. (2008) found that radar measurements from multiple sites and the corresponding NCEP reanalysis dataset indicated that a more unstable atmosphere tends to favor a maximum overlap over a random one. Indeed, we also find that an unstable atmospheric stratification over the TP and its corresponding weak wind shear often lead to a greater $\alpha$ when the layer distance is less than 4 km.

However, Figs. 4e and 4f also reveal an inconsistency in the relationship of layer distance and the corresponding wind shear and atmospheric stability. For example, we can see that the wind shear for a 2-km layer distance is greater than that for a 3-km distance, but the atmosphere is also more unstable. This difference is probably because two cloud layers with the same separation but different altitudes are sorted into the same statistical group.

Generally, the seasonal variation of $\alpha$ is correlated with the concurrent large-scale dynamics and atmospheric states.

Additionally, Fig. 5 shows the zonal variations of $\alpha$, $\partial\theta_{es}/\partial z$ and $dV/dz$ over the TP. Figs. 5a and 5d clearly indicate that $\alpha$ and the related layer distances are larger to the south of the TP and smaller to the north. Their zonal variations can be attributed to the

zonal differences of the atmospheric dynamics of the region (see Fig. 5b, 5c, 5e and 5f). Convective activity is more prevalent over the southern part of the TP than over the northern part (Fujinami and Yasunari, 2001). Thus, larger values of $\alpha$ can be found to the south. Based on the statistical results shown in Figs. 4 and 5, we confirm the importance of the impact of atmospheric dynamics on cloud overlap. Although $\alpha$ is

different than the multilayered cloud fraction that describes the ratio of the number of multilayered cloud profiles to the number of total sample profiles in a given grid box (Li et al., 2015), we found that the meteorological factors have similar effects on the





multi-layered cloud fractions. That is, an unstable atmosphere favors both a greater $\alpha$ and a greater multilayered cloud fraction, such that the percentage of multilayered cloud

systems is also higher over the southern part of the TP. Yuan et al. (2013) provided a detailed analysis of the global characteristics of the overlap of low and high altitude clouds through the use of combined active satellite datasets. These authors found that approximately 30% of the global distribution of low altitude clouds is overlapped by high altitude clouds, and the overlap rate (i.e., the multilayered cloud fraction) exhibits a

strong anti-correlation with the 700 hPa and 500 hPa vertical velocities over the ocean (with a correlation coefficient of -0.94). For comparison, we analyzed the seasonal and zonal variations of the 500 hPa vertical velocity over the TP (see the Figs. s1 and s2 in the supplementary material). As with the atmospheric stability, strong updrafts are more likely to occur over the southern part of TP (except the region north of 36 °N), especially

during the summer season, with the opposite seasonal and zonal changes as those associated with $\alpha$.

**3.2 Impacts of large-scale dynamics on the cloud overlap over the TP region**

In this section, we investigate the variations of $\alpha$ with changes in layer distance for different meteorological conditions to facilitate the parameterization of $\alpha$ for cases of

continuous clouds. Here, each meteorological factor over the TP region is grouped into one of four bins based on its values. In the present study, the four bins for $\partial\theta_{es}/\partial z$ are $\partial\theta_{es}/\partial z > 5$ K/km, $2 < \partial\theta_{es}/\partial z < 5$ K/km, $-1 < \partial\theta_{es}/\partial z < 2$ K/km and $\partial\theta_{es}/\partial z < -1$ K/km. For wind shear, the four bins are $dV/dz < 0.5$ m $\cdot$ s$^{-1}$/km, $0.5 < dV/dz < 2$ m $\cdot$ s$^{-1}$/km, $2 < dV/dz < 3.5$ m $\cdot$ s$^{-1}$/km and $dV/dz > 3.5$ m $\cdot$ s$^{-1}$/km. These groupings

ensure that a statistically significant number of samples fall within each bin (i.e., at least one hundred thousand samples per bin).

Fig. 6 illustrates the variations of $\alpha$ with layer distance for different large-scale atmospheric dynamics and different upper limit thresholds of cloud cover. When considering the proportion of cloud samples with small layer distances (<3.5 km), which

account for 92% of all samples with continuous clouds, we only provide the statistical results for layer distances smaller than 3.5 km. Given the upper limit (100%) of the cloud cover (Figs. 6a and 6c), the overlap properties of continuous clouds transition from a



maximum overlap at lower layer distances to a random overlap at greater distances. When the atmosphere is stable ($\partial\theta_{es}/\partial z > 5$ K/km), $\alpha$ becomes negative for the layer

distances of 1 km. When the atmosphere is unstable ($\partial\theta_{es}/\partial z < -1\,\text{K/km}$), this distance reaches 2 km. Given the same layer distances (e.g., 1.5 km), we observe differences of more than 0.5 in the $\alpha$ under unstable and stable atmospheric conditions. For discontinuous clouds, the variations of $\alpha$ with changes in layer distances do not follow the same inverse exponential model; however, the impact of atmospheric stability is obvious.

When the upper limit threshold of cloud cover is changed from 100% to 50% based on the suggestion from Tompkins and Di Giuseppe (2015), the sensitivity of $\alpha$ to changes in the large-scale atmospheric dynamics is significantly reduced, especially for changes in $\partial\theta_{es}/\partial z$ (Fig. 6b). Figs. 6b and 6d clearly show that $\alpha$ increases when the upper limit of cloud cover is reduced to 50%, especially given a stable atmosphere ($\partial\theta_{es}/\partial z > 5$ K/km).

It is possible that this increase results from the weakening of the dynamic factors on the scale of the cloud system for instances with lower cloud cover. As shown in Figs. 6b and 6d, the effects of the dynamic factors on $\alpha$ are obvious, although the sensitivity of $\alpha$ is reduced relative to that seen in Figs. 6a and 6c. Compared with $\partial\theta_{es}/\partial z$, the dependence of $\alpha$ on $dV/dz$ seems weak, especially in the presence of discontinuous

clouds. Naud et al. (2008) tested the sensitivity of $\alpha$ to wind shear at three sites and found that wind shear slightly affects $\alpha$ when the layer distance is larger than 2 km. However, in a recent study, Di Giuseppe and Tompkins (2015) demonstrated the important effect of wind shear on the global cloud overlap by using a combination of the CloudSat-CALIPSO cloud data and the ECMWF reanalysis dataset. These results and our

own results demonstrate that variations in cloud overlap are closely related to atmospheric dynamics, but their relationship is not stable and has spatial variations. Shortly, the effect of the atmospheric stability in cloud overlap may be more important over convective regions (e.g., the intertropical convergence zone and TP during summer season), while the effect of wind shear may be dominant over the mid-latitudes. Given

continuous cloud layers over the TP, Figs. 6b and 6d clearly illustrate that the effects of wind shear and atmospheric stability on cloud overlap should both be accounted for in the parameterization of $\alpha$ over the TP. In addition to these two factors, we also test the



sensitivity of $\alpha$ to the 500 hPa vertical velocity (see the Fig. s3 in the supplementary material). Although the seasonal cycle of $\alpha$ is related to the local vertical velocity, we

found that the sensitivity of $\alpha$ is rather weak for all vertical velocity bins. Naud et al. (2008) also found a weak sensitivity of $\alpha$ to vertical velocity, which they attributed to the uncertainty in the reanalysis dataset. Even when using the lower cloud cover threshold (50%), the proportion of negative $\alpha$ values is still high (increasing to 41% from the 39.8% for the cloud cover threshold of 100%) for cloud samples with smaller layer

distances (<3.5 km). Such a high proportion means that the contribution of negative $\alpha$ to the cloud overlap properties over the TP cannot be ignored in the parameterization of $\alpha$.

### 3.3 Treatment of negative overlap parameters and parameterization

As we know, the overlap parameter $\alpha$ for continuous cloud layers can be modeled as an inverse exponential function:

$$\alpha = e^{-D/L} \tag{6}$$

where $D$ is the layer separation distance and $L$ is the decorrelation length scale that characterizes the transition from the maximum to random overlap assumption. In previous studies, it is found that $L$ depends on the vertical and temporal resolutions of the lidar data or the spatial scale of the models, and the values of $L$ have varied across many

studies (Hogan and Illingworth, 2000; Mace and Benson-Troth, 2002; Pincus et al., 2005). For example, Barker (2008) found a wide range of $L$ values, with a median value of 2 km, when analyzing the global CloudSat and CALIPSO datasets. Oreopoulos and Norris (2011) derived $L$ based on radar measurement taken over the Southern Great Plains (SGP) of the USA. Their results indicated that the value of $L$ ranges from 2 to 4.5 km across

different seasons and that smaller spatial scales correspond with smaller $L$ values. In other studies, $L$ was usually a function of latitude or total cloud cover (Shonk et al., 2010; 2014; Yoo et al., 2014). In our study, however, an obvious drawback of Eq. (6) is that the negative values of $\alpha$ cannot be parameterized in the inverse exponential modeling. Oreopoulos and Norris (2011) also noted that the negative values of $\alpha$ in their dataset

were too frequent for the use of inverse exponential modeling to be robust.

Our dataset show that the proportion of negative $\alpha$ values decreases as the averaged cloud height of the two cloud layers increases, except for clouds with a mean height exceeding 15 km (Fig. 7). Thus, the overlap parameter for the lower atmosphere of the





TP is more likely to be negative, especially for low altitude clouds with greater layer
distances. Overall, the proportion of negative $\alpha$ values increases from approximately 15%
to 80% as the layer distance increases from 1 km to 4 km, with a mean proportion of
approximately 41%. Oreopoulos and Norris (2011) suggested that the inverse exponential
function is still valid for negative $\alpha$ values when the impact of replacing these negative
values with zeros is small. In view of the observed total cloud cover must be between
predicted cloud covers by minimum and maximum overlap assumptions, we propose
another method of quantifying the degree of cloud overlap through the linear combination
of the maximum and minimum overlap assumptions, forming a new "overlap parameter"
$\alpha_{new}$:

$$C_{i,j}^{obs} = \alpha_{new} C_{i,j}^{max} + (1 - \alpha_{new}) C_{i,j}^{min} \tag{7}$$

As with Eq. (5), $\alpha_{new}$ can also be written as:

$$\alpha_{new} = \frac{C_{i,j}^{obs} - C_{i,j}^{min}}{C_{i,j}^{max} - C_{i,j}^{min}} = e^{-D/L_1} \tag{8}$$

As $\alpha_{new}$ is always larger than 0, Eq. (8) allows its simple parameterization through
inverse exponential modeling. Table 1 lists several parameterization schemes for the
decorrelation length scale $L$. Schemes 4, 5 and 6 are derived by considering $L$ as a
function of wind shear or atmospheric stability based on both a multiple linear regression
method and our statistical dataset. The regression formula of $L$ and $L_1$ can be written as:

$$L = L_\alpha - b1 \frac{d\theta_{es}}{dz} - b2 \frac{dV}{dz}$$
$$or \tag{9}$$
$$L_1 = L_{\alpha 1} - c1 \frac{d\theta_{es}}{dz} - c2 \frac{dV}{dz}$$

Here, $L_\alpha$, $L_{\alpha 1}$, $b1$, $b2$, $c1$ and $c2$ are the fitting parameters. In Table 1, scheme 7 is the
previously mentioned scheme from Di Giuseppe and Tompkins (2015), i.e., using the
global CloudSat-CALIPSO cloud data and ECMWF reanalysis dataset to parameterize $L$
as a function of wind shear. The method of Shonk et al. (2010) was also used, which
proposed an empirical linear relationship between $L$ and latitude (see scheme 8). For
detailed information about each scheme, see the descriptions in Table 1. Note that the
R-squared values ($R^2$) for schemes 4, 5 and 6 are 0.98, 0.44 and 0.94, respectively.



Fig. 8 presents the zonal differences between the calculated and observed cloud covers for the aforementioned schemes and indicates that the maximum (scheme 2) and minimum overlap assumptions (scheme 3) result in large cloud cover biases at most of the layer distances, especially for layer distances greater than 1 km and less than 2 km (where the bias exceeds 5%). The bias of the random overlap assumption (scheme 1) is

greater (>5%) when the layer distance is less than 1 km, although a lower bias may be found for greater distances. As in scheme 2 (the maximum overlap assumption), scheme 7 significantly underestimates the cloud cover for layer distances above 1 km, whereas the obvious underestimation (e.g., >5%) of scheme 8 only exists for layer distances between 2 and 3 km. Notably, the large bias of scheme 7 may occur because the scheme is based

on the global CloudSat-CALIPSO measurements and ECMWF reanalysis dataset for a short period (January-July 2008); as such, some obvious regional or seasonal cloud overlap properties are easily obscured by global averaging. Another possible cause is that the other dynamic factors (e.g., the atmospheric stability) were not considered in the parameterization of this scheme. If we replace the negative $\alpha$ values with zero and

parameterize $L$ as a function of wind shear and atmospheric stability (see scheme 4), scheme 4 still results in some underestimations, especially for layer distances between 2 and 3 km, as seen in Fig. 8. However, when we consider the true cloud overlap as the result of the linear combination of the maximum and minimum overlaps (schemes 5 and 6) and the influence of wind shear, the bias of scheme 5 is reduced to $\pm 2\%$. In

comparison, scheme 6 has a relatively lower bias at large layer distances but higher R-squared values ($R^2$=0.94). These results indicate that the effects of both wind shear and atmospheric stability on cloud overlap should be accounted for in the parameterization of the overlap parameter $\alpha$ over the TP, and the use of Eq. (8) can considerably reduce the cloud cover bias caused by the negative $\alpha$ values.

**4.   Conclusions and discussion**

The "roof of the world", i.e., the Tibetan Plateau, has experienced obvious climate changes over the past three decades, and these climate changes are partially related to the cloud variations over the TP region. Vertical cloud overlap, which is one of the most important macro-physical properties of clouds, is particularly significant for predicting

the total cloud cover and for the calculations related to the radiative budget in weather





forecasting and climate models. However, the study of cloud overlap over the TP region has received less attention than necessary because of the limited regional cloud observations and our incomplete knowledge of the physical processes of clouds. In this study, we collect 4 years (2007–2010) of data from the CloudSat 2B-GEOPROF-LIDAR dataset, the ECMWF-AUX dataset and ERA-Interim daily 6-hourly reanalysis to analyze the seasonal and zonal variations of cloud overlap over the TP region and to further evaluate the effects of atmospheric dynamics on cloud overlap.

Our results show that a random overlap assumption slightly underestimates the total cloud coverage of discontinuous cloud layers over the TP, whereas the overlap parameter for continuous cloud layers has sharply transitions from a maximum to random overlap with an increase of cloud layer distance, eventually trending towards a minimal overlap (e.g., negative $\alpha$ values) as the cloud layer separation distance exceeds 1.5 km. As seen in previous studies (Hogan and Illingworth, 2000; Mace at al., 2002; Naud et al., 2008; Oreopoulos and Norris, 2011), the overlap parameter $\alpha$ over the TP is sensitive to the spatial sampling scale. However, recent research has attributed some of these phenomena (e.g., the negative $\alpha$ values and sampling scale-dependence) to data truncation errors, which result from using a sampling scale smaller than the cloud systems. To reduce the errors in $\alpha$ and its corresponding decorrelation length, we followed the suggestion of Tompkins and Di Giuseppe (2015) and applied a simple filter for the instances of continuous cloud layers, retaining only the cloud samples with an upper limit of cloud cover less than 50%. The proportion of negative $\alpha$ values was still high (approximately 41%) despite the given cloud fraction threshold (99%), spatial sampling scale (50 km) and the lower limit (50%) of cloud cover. Compared with the globally averaged cloud overlap characteristics, we propose that the high proportion of negative $\alpha$ may be associated with the unique topographical forcing and thermo-dynamical environment of the TP.

By analyzing the atmospheric dynamics over the TP, we found that the seasonal and zonal variations of $\alpha$ are related to variations of several meteorological factors, and the sensitivities of $\alpha$ to wind shear and atmospheric stability are well above that of the vertical velocity at 500 hPa. Generally, the unstable atmospheric stratification and correspondingly weak wind shear over the TP lead to greater $\alpha$ values, whereas a stable





atmosphere tends to result in a negative $\alpha$. As such, the effects of wind shear and atmospheric stability on cloud overlap should both be considered in the parameterization of $\alpha$ over the TP. The inverse exponential modeling of $\alpha$ is not always robust owing

to the frequent existence of negative $\alpha$ over the TP. As a result, we propose a viable scheme for quantifying the degree of cloud overlap through a linear combination of the maximum and minimum overlaps, and further parameterize $L$ as a function of the wind shear and atmospheric stability. Compared with other parameterizations, this new scheme considerably reduces the bias in the predicted cloud coverage over the TP. Although the

parameterization method derived in our study focuses on only the TP and may therefore have some limitations for global applications, our results suggest that the parameterization of the decorrelation length scale $L$ through the related physical processes and the consideration of multiple dynamic factors and microphysical effects (e.g., precipitation) has the potential to improve simulations of the total cloud cover in

models.

     Previous studies have pointed out that precipitation may bias the cloud overlap statistics toward maximum overlap (Mace et al., 2009; Di Giuseppe and Tompkins). However, the present study doesn't eliminate the influence of precipitation on the overlap parameter, which means that overlap parameter $\alpha$ will be smaller (or more negative) if

cloud samples with precipitation are removed from our analysis. In addition, the future work would be to use the regional climate models to evaluate the impact of new scheme on the radiative budget compared with existing overlap parameterization in models. Cloud overlap parameter is significantly controlled by atmospheric dynamics, therefore the long-term variations of meteorological factors are bound to affect the trend of cloud

overlap and the following total cloud cover and radiation budget. Indeed, a recent study has shown that rapid warming and an increase of atmospheric instability over the TP leads to more frequent deep clouds, which are responsible to the reduction of solar radiation over the TP (Yang et al., 2012). In view of the decreasing trend of total cloud cover over this region (You et al., 2014), we may infer that the degree of cloud overlap

over the TP possibly has an increasing trend. However, as stated by the Tompkins and Di Giuseppe (2015), true cloud overlap parameter will be biased at a given spatial sampling scale if one doesn't consider the cloud system size. Thus whether the trend is true or



significant over different parts of TP still needs further quantified because related cloud system scales over these regions are possibly changed as well.


*Competing interests.* The authors declare that they have no conflict of interest.

*Acknowledgements.* This research was jointly supported by the Foundation for Innovative Research Groups of the National Science Foundation of China (grant no. 550 41521004) and key Program of the National Natural Science Foundation of China (41430425), National Science Foundation of China (grant nos. 41575015 and 41575143) and the China 111 project (grant no. B13045). We would like to thank the CALIPSO, CloudSat and ERA-Interim science teams for providing excellent and accessible data products that made this study possible.

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



Table 1. Several parameterizations of decorrelation length scale $L$ from the exponential fit as a function of atmospheric stability $\partial \theta_{es}/\partial z$, wind shear $dV/dz$ or latitude $\Phi$

| Scheme | description | decorrelation length scale $L$ |
|---|---|---|
| Scheme 1 | Random | |
| Scheme2 | Maximum | |
| Scheme 3 | Minimum | |
| Scheme 4[1] | Random/Maximum | $L = 1.55 - 0.03 \times \dfrac{d\theta_{es}}{dz} - 0.06 \times \dfrac{dV}{dz}$ |
| Scheme 5[2] | Minimum/Maximum | $L = 1.73 - 0.08 \times \dfrac{dV}{dz}$ |
| Scheme 6[2] | Minimum/Maximum | $L = 1.92 - 0.05 \times \dfrac{d\theta_{es}}{dz} - 0.08 \times \dfrac{dV}{dz}$ |
| Scheme 7[3] | Random/Maximum (Di Giuseppe and Tompkins, 2015) | $L = 4.4 - 0.45 \times \dfrac{dV}{dz}$ |
| Scheme 8[4] | Random/Maximum (Shonk et al., 2010) | $L = 2.899 - 0.02759 \times \lvert \Phi \rvert$ |

[1]Scheme 4 is based on Eq. (4), but replaces the negative $\alpha$ with 0; Meantime, decorrelation length scale $L$ is parameterized as a function of atmospheric stability and wind shear; [2]Scheme 5 and 6 both are based on Eq. (7), and decorrelation length scale $L$ is parameterized as a function of atmospheric stability or wind shear; [3]Scheme 7 is based on Eq. (4), and decorrelation length scale $L$ is parameterized as a function
of wind shear; [4]Scheme 8 is based on Eq. (4), and decorrelation length scale $L$ is parameterized as a function of latitude; (please see section 3.3)

**Figure captions**

Figure 1. (a) CloudSat overpass tracks (blue line: daytime; red line: nighttime) over the Tibetan Plateau (27°N-39°N; 78°E-103°E); (b) A sample of CloudSat 2B-GEOPROF-LIDAR cloud mask product along the ground track of 200km (white color: cloud fraction>99%; light blue: 0<cloud fraction<99%; deep blue: clear sky; orange color: surface).


Figure 2. The sensitivity of cloud overlap parameter $\alpha$ to cloud fraction threshold for non-continuous (left) and continuous (right) cloud pairs on a 50 km spatial sampling scale.

Figure 3. (a), (c) The sensitivity of $\alpha$ to the spatial sampling scale for non-continuous and continuous cloud pairs; (b) The probability distribution functions (PDFs) of the along-track horizontal scales of cloud system at different height over TP region; (d) Cloud sample numbers for the non-continuous and continuous clouds at a given sampling





scale of 50km. The percentages represent the proportions of cloud sample below

corresponding layer distance to all samples.

Figure 4. (a),(b) and (c) The seasonal variations of the pentad-averaged cloud overlap parameter $\alpha$, degree of conditional instability to moist convection $\partial\theta_{es}/\partial z$ and wind shear $dV/dz$ for the continuous clouds over the TP ; (d), (e) and (f) The seasonal

variations of the pentad-averaged $\alpha$, $\partial\theta_{es}/\partial z$ and $dV/dz$ for the continuous clouds at given layer distances (red: 2km; black: 3km).

Figure 5. (a),(b) and (c) The zonal variations of the $\alpha$, $\partial\theta_{es}/\partial z$ and wind shear $dV/dz$ for the continuous clouds over the TP; (d), (e) and (f) The zonal variations of the $\alpha$,

$\partial\theta_{es}/\partial z$ and $dV/dz$ for the continuous clouds at given layer distances (red: 2km; black: 3km).

Figure 6. The variation of overlap parameter $\alpha$ with layer distance under different large-scale dynamics and upper limit thresholds of cloud cover (100%) for (a) and (c);

(50% cloud cover threshold) for (b) and (d).

Figure 7. The proportions of negative $\alpha$ at the different layer distances and averaged cloud heights of two cloud layers for the continuous clouds.

Figure 8. The zonal difference of cloud cover between calculated and observed for different schemes (see the Table 1) and its variation with layer distance.








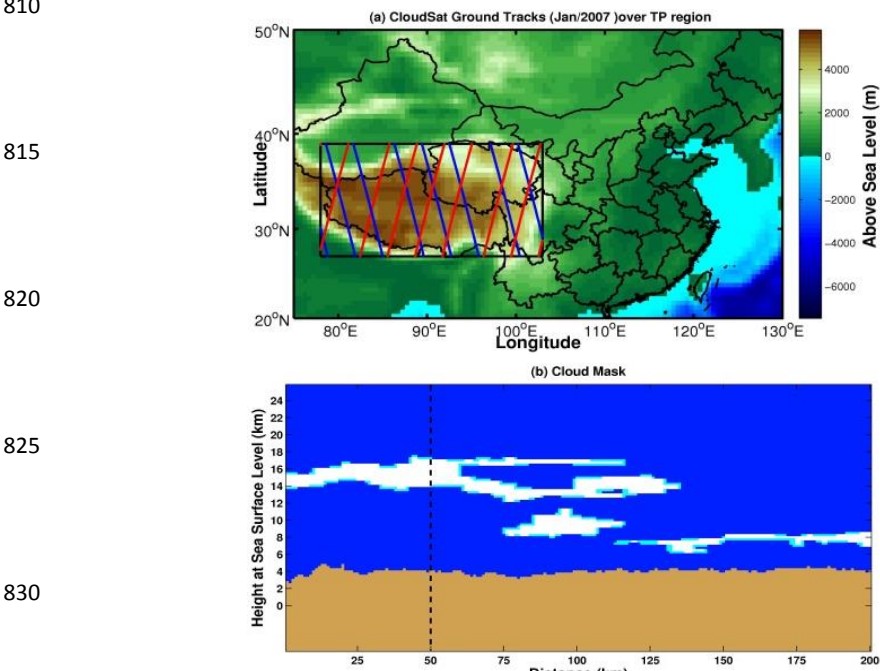

Figure 1. (a) CloudSat overpass tracks (blue line: daytime; red line: nighttime) over the Tibetan Plateau (27 °N-39 °N; 78 °E-103 °E); (b) A sample of CloudSat 2B-GEOPROF-LIDAR cloud mask product along the ground track of 200km (white color: cloud fraction>99%; light blue: 0<cloud fraction<99%; deep blue: clear sky; orange color: surface).






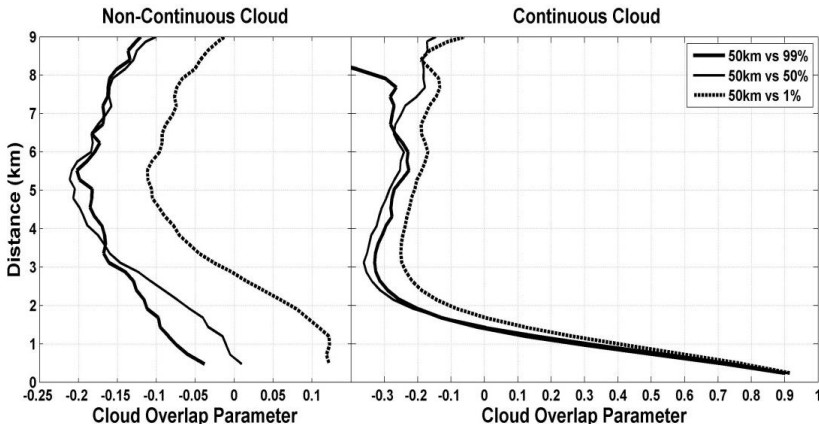

Figure 2. The sensitivity of cloud overlap parameter $\alpha$ to cloud fraction threshold for non-continuous (left) and continuous (right) cloud pairs on a 50 km spatial sampling scale.

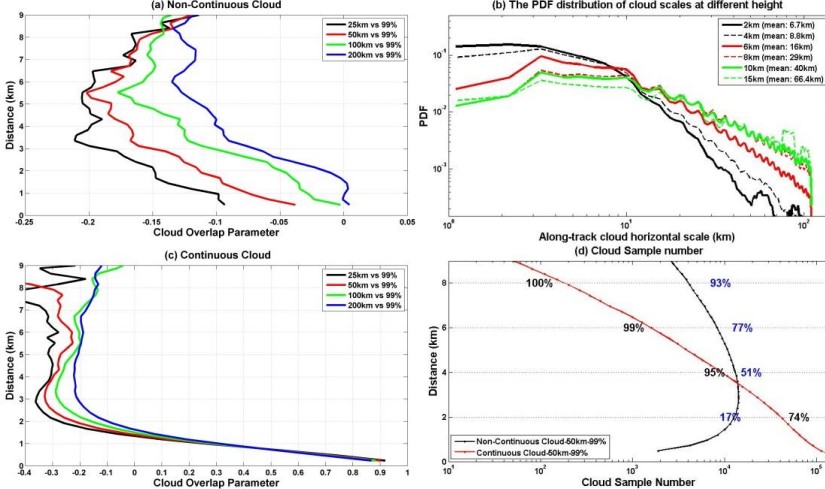

Figure 3. (a), (c) The sensitivity of $\alpha$ to the spatial sampling scale for non-continuous and continuous cloud pairs; (b) The probability distribution functions (PDFs) of the along-track horizontal scales of cloud system at different height over TP region; (d) Cloud sample numbers for the non-continuous and continuous clouds at a given sampling scale of 50km. The percentages represent the proportions of cloud sample below corresponding layer distance to all samples.





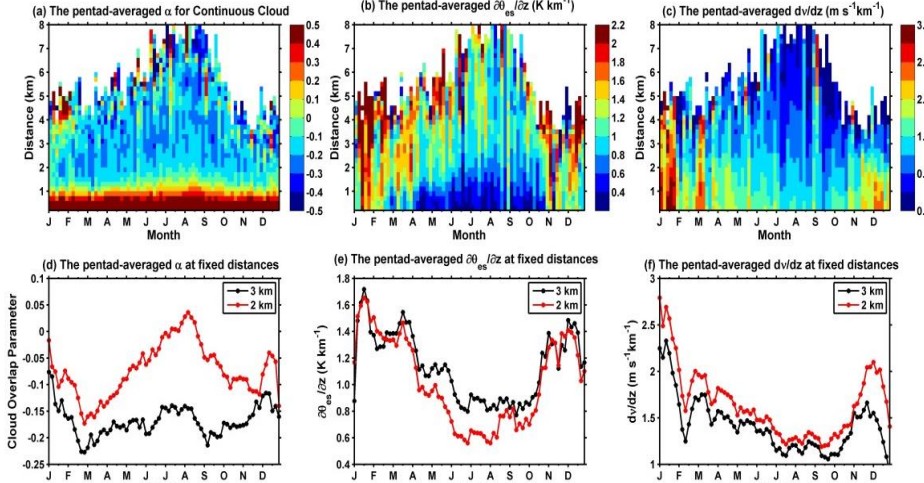

Figure 4. (a),(b) and (c) The seasonal variations of the pentad-averaged cloud overlap parameter $\alpha$, degree of conditional instability to moist convection $\partial\theta_{es}/\partial z$ and wind shear $dV/dz$ for the continuous clouds over the TP ; (d), (e) and (f) The seasonal variations of the pentad-averaged $\alpha$, $\partial\theta_{es}/\partial z$ and $dV/dz$ for the continuous clouds at given layer distances (red: 2km; black: 3km).



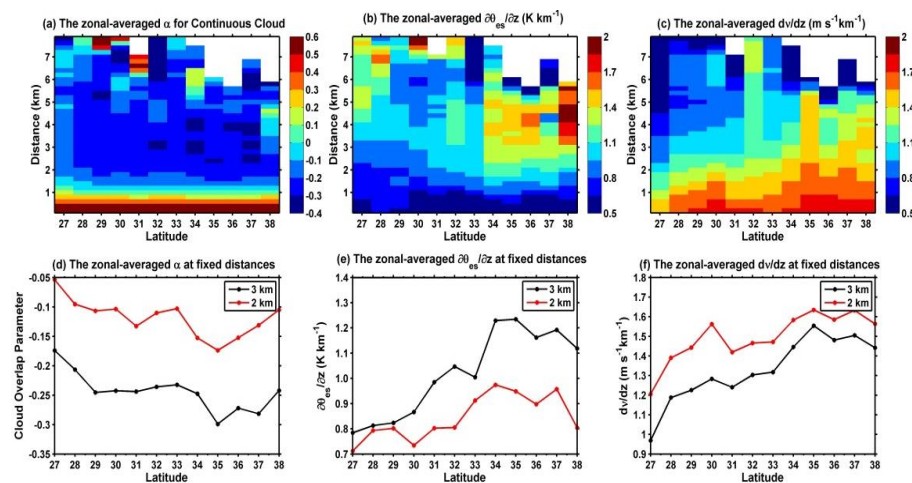

Figure 5. (a),(b) and (c) The zonal variations of the $\alpha$, $\partial\theta_{es}/\partial z$ and wind shear $dV/dz$ for the continuous clouds over the TP; (d), (e) and (f) The zonal variations of the $\alpha$, $\partial\theta_{es}/\partial z$ and $dV/dz$ for the continuous clouds at given layer distances (red: 2km; black: 3km).




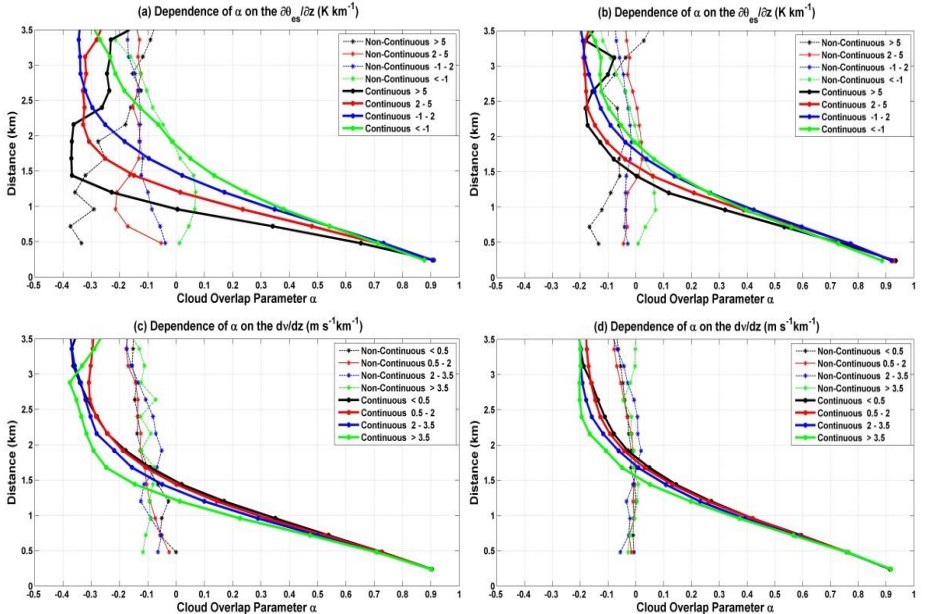

Figure 6. The variation of overlap parameter $\alpha$ with layer distance under different large-scale dynamics and upper limit thresholds of cloud cover (100%) for (a) and (c); (50% cloud cover threshold) for (b) and (d).


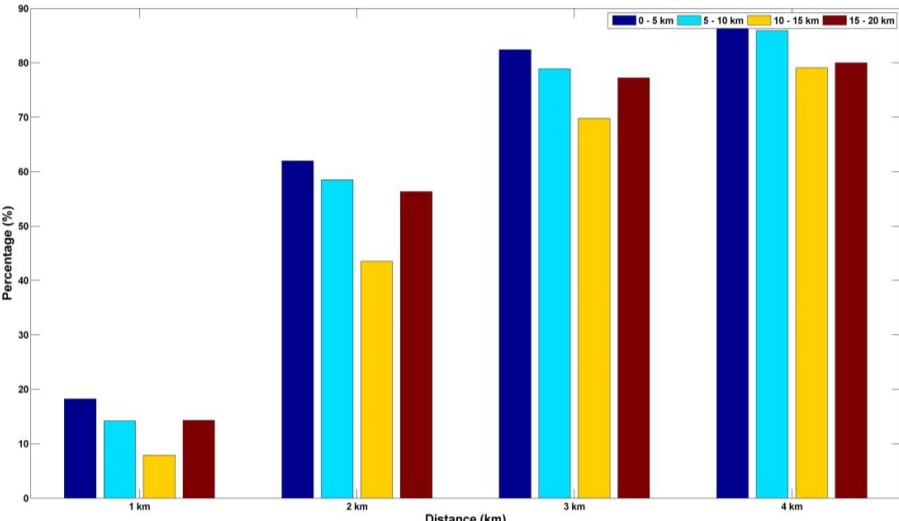

Figure 7. The proportions of negative $\alpha$ at the different layer distances and averaged cloud heights of two cloud layers for the continuous clouds.




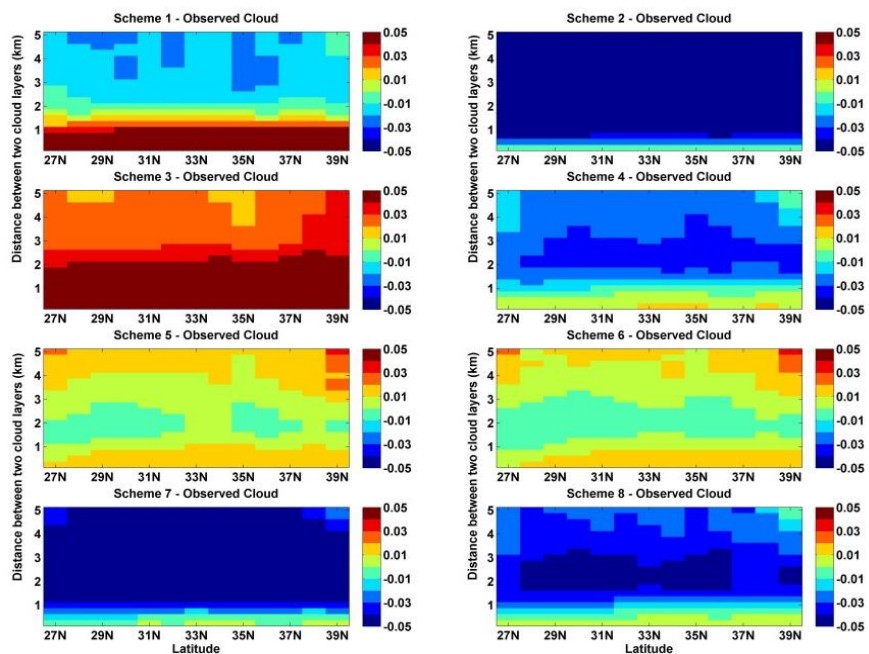

Figure 8. The zonal difference of cloud cover between calculated and observed for different schemes (see the Table 1) and its variation with layer distance.
