# Peer review of "Figure S1. (a) and (b) The seasonal variations of the pentad-averaged cloud overlap parameter $\alpha$ , and vertical velocity at 500hPa for the continuous clouds over the TP ; (c) and (d) The seasonal variations of the pentad-averaged $\alpha$ and vertical velocity for the continuous clouds at g"

_Atmospheric Chemistry and Physics, 2017_

## Referee Comment (RC1) · Anonymous Referee #1 · 30 Aug 2017

This is a very short review because there is a problem at the onset: the data product that is used to examine cloud overlap is said to be the "CloudFraction" product from the GEOPROF-LIDAR files. No other cloud product/data field is mentioned. In Mace et al., 2009 it is explained that this is the lidar only cloud fraction calculated when the lidar cloud detections are matched to the Cloudsat bins. Consequently, it is subjected to large attenuation in thick clouds or in conditions of multiple cloud layers in a column. When in section 2.3 the authors find a tendency for the discontinuous cloud layers to exhibit a minimum cloud overlap, this is expected when the lidar alone is used: the low level clouds will be more readily detected if there are no high clouds, so in situations of minimum overlap. And the same can be said in continuous cloud layer situations. So at this point none of the conclusions concerning specifically the Tibetan Plateau clouds

have any physical bearing other than variations in their optical thickness that pushed the overlap estimate towards minimum. What I would strongly advise the authors to do is to use the cloud base and top heights given in the "LayerBase" and "LayerTop" fields of the GEOPROF-LIDAR files and construct the cloud mask with these quantities. Then redo the whole study and see how different the results are.

When going back to the Di Giuseppe and Tompkins (2015) paper that the author refer to, this is clearly explain: they use either the radar alone or the radar with the additional information given by the "cloudfraction" field, not the latter alone.

Now, if I am mistaken and the dataset used here is in fact a combination of the "cloud-Fraction" product with some other radar information, then this is not clear and should be specified. However, in view of the results I have doubts that this is the case.

So at this point, I recommend major revisions to include radar information and redo the analysis. I do not think that it will take a lot of time, the analysis seems rather well presented and constructed, and the authors have already the data files with the relevant information.

---

## Referee Comment (RC2) · Anonymous Referee #2 · 1 Sep 2017

This paper looks at cloud overlap over the Tibetan Plateau. Studies have calculated cloud overlap globally before, although resulting parameterisations have not been valid for clouds over the Tibetan Plateau, so the authors use satellite radar and lidar data to build up two-dimensional slices of cloud and evaluate the overlap parameter (based on the description of Hogan and Illingworth, 2000). They also use ECMWF data to describe the meteorological conditions and derive along-track wind shear and conditional instability and investigate the local effect of overlap parameter on these quantities. They find overlap here to generally be minimum for discontinuous clouds and distant layers in continuous clouds, and define a new overlap parameter that varies between minimum and maximum overlap. The decorrelation scale of this parameter is then regressed onto wind shear and conditional instability to produce a new equation for

overlap as a function of these quantities over the Tibetan Plateau that performs better than many standard methods.

(I should point out at this point that I wrote much of this before reading the comments from Reviewer #1 and, if indeed the authors are using the wrong variable, some of the following comments may well not be relevant after changes according to this.)

Generally, I thought the main points of the paper were interesting, and the authors' new equation for decorrelation scale is a useful result. However, the presentation of the work needs a great deal of attention. This paper came across as a list of everything the authors did, with a description of every result, yet little scientific insight as to the implications of what they found. The result of this was that the reader is faced with through the details of several minor results (in particular those presented at the end of section 2) before reaching the key results in the later parts of section 3.

The highlight of the paper for me was the new overlap parameterisation and the comparison with the other overlap methods, which was presented in section 3.3. To draw more attention to this, I strongly recommend elaborating on the detail of this part of the experiment and heavily trimming some of the less important early results (again, those at the end of section 2). I would also recommend tidying up the results section. The authors tend to provide a great deal of information about what a figure shows, but significantly less on its interpretation.

In summary, I think the results are interesting (although the authors should definitely check the comments from Reviewer #1), but the text needs to be heavily edited and restructured before this paper is suitable for publication.

Here are a few specific comments I had while reading.

At a few points throughout the paper, there are allusions to the climate changes experienced by the Tibetan Plateau, but little detail on what these changes are, and indeed how misrepresentation of cloud overlap could affect predictions of these changes.

Line 73: overlap parameterisations, rather than overlap states. They are not really states that occur in the atmosphere, but parameterisations that we apply to them in models.

You alternate throughout between cloud cover and cloud coverage. Cloud cover is more commonly used.

Lines 75 onwards: you introduce maximum, minimum and random overlap, but do not state exactly how they are calculated. Describing the cloud cover for two layers under each overlap assumption is useful, but a proper physical definition of what the overlap schemes mean will help. (This could also belong in the data/methods section.)

Line 88. "Accurate" perhaps isn't the word you mean – this implies that the radar always calculates the exact masks of the true clouds.

Line 94: "space-based" is better than "space-borne" (the satellite is not carried along by space!).

Lines 135 onwards. This section was not clear, and I needed to read it multiple times to get an idea of what you meant. For a start, I assume that the CloudSat profiles are columns of zeros and ones for clear and cloudy, which are then combined to give two-dimensional scenes? Are CloudSat bins the same as CloudSat profiles (presumably the bins are height layers within the profiles)? And what are the horizontal and vertical resolutions referring to? This all needs to be made clear or the reader will struggle to follow.

Section 2.3. The first time I read this, again it was not clear here why and how is the domain divided into different domain sizes. You should probably describe this more explicitly. I assume the data is simply divided up into sections of these different lengths? You also define two terms ("spatial sampling size" and "domain size") for this quantity, but then proceed calling it something else ("spatial sampling scale").

Line 205. Going back to my main points above – I don't understand why this result is

in the paper in this much detail. Figure 2 shows that there is little difference in terms of overlap statistics whether the threshold is 99% or 50%, but I don't see how this particularly justifies the use of the 99% threshold. I wonder if this result is even worth a mention – removing it would mean that the definition of the threshold is no longer necessary.

Several of your paragraphs are far too long. A reader will be daunted by paragraphs that span multiple pages. I recommend breaking these paragraphs down into manageable sections that describe one element of the study.

Figure 3. Putting panel (c) to the top right would match the position of the equivalent panel in Figure 2.

Line 235 to end of paragraph. I am not sure what you mean here – I think it is about the domain size needing to be greater than the cloud scales at each level. But now there is another threshold that is introduced of 50% that is different in definition to the previous threshold. Basically, as far as I can tell, the overview of the latter parts of section 2.3 is to pick a set of values for the two thresholds and the spatial sampling scale. This should probably be partitioned off into a separate section.

Sections 3.1 and 3.2. These were very difficult to read and need reworking – the material within them is fine, but the key results need to be emphasised and worked into more of a scientific story.

Table 1 and Figure 8: which of the overlap schemes in the table are yours? By the looks of it, it's 5 and 6, but it wasn't immediately obvious. Why not give the schemes names that highlight those that are yours, then when you compare the performance of the schemes over the Tibetan Plateau, it is easier to see that yours perform best.

Figure 8. This figure would be clearer if the colour bar used white for zero. Then it would be clearer which overlap schemes produce biases.

Section 4. Your conclusion section is just a summary of the results and some future

ideas, with little extra insight. You need to place your results here into context. You allude at many points along the way to climate change and modelling over the Tibetan Plateau and how radiation budget is affected by cloud overlap issues here – how could this parameterisation help?
* * *

---

## Author Comment (AC1) · 7 Nov 2017

**Response to Reviewer #1's Comments:**

**Jiming Li et al. (Author)**

**We are very thanks Review #1 for pointing out the vital weaknesses of this paper and addressing significant comments on the original manuscript, which are very helpful and have led to significant improvements of this paper. Indeed, too many thick clouds were missed in our previous results. By combining the information of "LayerBase", "LayerTop" heights and "CloudFraction" parameter, we reconstruct the cloud mask and redo the whole study. As indicated by Review #1, the results are very different (see the revised paper). In addition, some superfluous information in each section was deleted and some interpretations in each section were added in order to make the manuscript more clear. Some grammatical errors already were corrected in the revision and the paper also be edited by a native English speaker to make it more readable.**

---

## Author Comment (AC2) · 7 Nov 2017

**Response to Reviewer #2's Comments:**

**Jiming Li et al. (Author)**

**We are very grateful for the Review #2's detailed comments and suggestions, which help us improve this paper significantly. In the original manuscript, we are sorry to only use cloud fraction information from lidar so that our results missed too many thick clouds. By combining the information of "LayerBase", "LayerTop" heights and "CloudFraction" parameter, we reconstruct the cloud mask and redo the whole study. Based on two Reviewers' comments, we reorganized the manuscript and paid more attentions to interpret our finding and results. In addition, some superfluous information in each section was deleted and some interpretations in each section were added in order to make the manuscript more clear.**

**Specific responses**

(1) At a few points throughout the paper, there are allusions to the climate changes experienced by the Tibetan Plateau, but little detail on what these changes are, and indeed how misrepresentation of cloud overlap could affect predictions of these changes.

**Response:** We agreed with reviewer. In the revised paper, we added some information and interpretations of climate changes over TP region (see Line 62-70 in Introduction section).

"**For example**, many studies have showed that significant warming occurs in the TP region during the last decades and will continue to warm in the future (e.g., Duan et al., 2006; Wang et al., 2008). The rapid warming has caused glacier retreat and expansion of glacier-fed lakes (Zhu et al., 2010), permafrost degradation and temperature increasing (Cheng and Wu, 2007), heating source became weakened (Yang et al., 2011) and corresponding variation of summer precipitation downstream (Duan et al., 2013). In addition to increased greenhouse gas emission, Kang et al. (2010) summarized that the changes of cloud cover also is one of dominant factors causing the rapid warming over TP region. Indeed, some studies have linked the rapid warming to variations in the cloud

cover over the TP region (e.g., Chen and Liu, 2005; Duan and Wu, 2006; Li et al., 2006; Yang et al., 2012; You et al., 2014; Wu et al., 2014). Such as, a recent study has indicated that increased nocturnal cloud cover over the northern TP would warm the nighttime temperature via enhanced atmospheric back-radiation, while decreased daytime cloud over the southern TP has contributed to the increasing of surface air temperature during daytime(Duan and Xiao, 2015). Based on the above studies, it is necessary to reasonably simulate the cloud cover in the climate models in order to better predictions these climate changes over TP."

(2) Line 73: overlap parameterizations, rather than overlap states. They are not really states that occur in the atmosphere, but parameterizations that we apply to them in models. You alternate throughout between cloud cover and cloud coverage. Cloud cover is more commonly used.
**Response:** In the revised paper, we already corrected it. Thanks reviewer.

(3) Lines 75 onwards: you introduce maximum, minimum and random overlap, but do not state exactly how they are calculated. Describing the cloud cover for two layers under each overlap assumption is useful, but a proper physical definition of what the overlap schemes mean will help. (This could also belong in the data/methods section.)
**Response:** In the revised paper, we added some interpretations to each overlap assumption (see Line 85-90). In addition, the physical definitions were emphasized in the section 2.3 (see Line 220).
"**The maximum assumption** minimizes the total cloud cover, while minimum assumption produces minimally overlap between cloud layers and results in maximum total cloud cover. The total cloud cover predicted by the random assumption will fall somewhere between maximum and minimum assumptions. For example, if the cloud covers in two model layers are given as 50%, then the maximum overlap will result in a total cloud cover of 50%, and a minimum overlap will result in an overcast condition (a complete cloud cover, i.e., 100%)"

(4) Line 88. "Accurate" perhaps isn't the word you mean – this implies that the radar

always calculates the exact masks of the true clouds.

**Response:** It was corrected in the revised manuscript.

(5) Line 94: "space-based" is better than "space-borne" (the satellite is not carried along by space!).

**Response:** It was corrected in the revised manuscript.

(6) Lines 135 onwards. This section was not clear, and I needed to read it multiple times to get an idea of what you meant. For a start, I assume that the CloudSat profiles are columns of zeros and ones for clear and cloudy, which are then combined to give two dimensional scenes? Are CloudSat bins the same as CloudSat profiles (presumably the bins are height layers within the profiles)? And what are the horizontal and vertical resolutions referring to? This all needs to be made clear or the reader will struggle to follow.

**Response:** Related information was added in the section 2.1 to make the manuscript more clear.

"**In the dataset,** every CloudSat profile includes 125 height layers (e.g., vertical bin), and the "*CloudFraction*" parameter reports the fraction of the lidar volume within each radar vertical bin that contains hydrometeors (Mace et al., 2009; Mace and Zhang, 2014). Several previous studies have identified a cloudy atmospheric bin based on different thresholds of the lidar-identified cloud fraction, including a 99% (Barker, 2008; Di Giuseppe and Tompkins, 2015) or 50% threshold (Haladay and Stephens, 2009; Verlinden et al., 2011). Here, a threshold of 99% is used in our study. However, due to the significant attenuation of lidar signals to the optically thick layers, this parameter fails to provide the "*CloudFraction*" value in those optically thick layers. Thus, we have to use other radar information (that is, cloud "*LayerBase*" and "*LayerTop*" fields) from the aforementioned dataset as supplementary to construct the complete two-dimensional cloud mask (See Fig. 1b)".

(7) Section 2.3. The first time I read this, again it was not clear here why and how is the domain divided into different domain sizes. You should probably describe this more

explicitly. I assume the data is simply divided up into sections of these different lengths? You also define two terms ("spatial sampling size" and "domain size") for this quantity, but then proceed calling it something else ("spatial sampling scale").

**Response:** We agreed with reviewer. In the revised paper, we already added some information and interpretations in the section 2.3.

(8) Line 205. Going back to my main points above – I don't understand why this result is in the paper in this much detail. Figure 2 shows that there is little difference in terms of overlap statistics whether the threshold is 99% or 50%, but I don't see how this particularly justifies the use of the 99% threshold. I wonder if this result is even worth a mention – removing it would mean that the definition of the threshold is no longer necessary.

**Response:** We agreed with reviewer. In the revised paper, this result is already deleted.

(9) Several of your paragraphs are far too long. A reader will be daunted by paragraphs that span multiple pages. I recommend breaking these paragraphs down into manageable sections that describe one element of the study.

**Response:** We already changed it in the revised paper.

(10) Line 235 to end of paragraph. I am not sure what you mean here – I think it is about the domain size needing to be greater than the cloud scales at each level. But now there is another threshold that is introduced of 50% that is different in definition to the previous threshold. Basically, as far as I can tell, the overview of the latter parts of section 2.3 is to pick a set of values for the two thresholds and the spatial sampling scale. This should probably be partitioned off into a separate section.

**Response:** We agreed with reviewer. In the revised paper, we already added a section to interpret the selection of thresholds for cloud cover and spatial scale (see the section 2.4).

(11) Sections 3.1 and 3.2. These were very difficult to read and need reworking – the material within them is fine, but the key results need to be emphasized and worked into more of a scientific story.

**Response:** We agreed with reviewer. In the revised paper, we already reorganized these sections and added some necessary explanations (see section 3.1 and 3.2).

(12) Table 1 and Figure 8: which of the overlap schemes in the table are yours? By the looks of it, it's 5 and 6, but it wasn't immediately obvious. Why not give the schemes names that highlight those that are yours, then when you compare the performance of the schemes over the Tibetan Plateau, it is easier to see that yours perform best.

**Response:** It was corrected in the revised manuscript. (See Figs.6 and 7, table 1).

(13) Figure 8. This figure would be clearer if the colour bar used white for zero. Then it would be clearer which overlap schemes produce biases.

**Response:** It was corrected in the revised manuscript. (See Figs.6 and 7).

(14) Section 4. Your conclusion section is just a summary of the results and some future ideas, with little extra insight. You need to place your results here into context. You allude at many points along the way to climate change and modeling over the Tibetan Plateau and how radiation budget is affected by cloud overlap issues here – how could this parameterization help?

**Response:** We agreed with reviewer. In the revised paper, we tidied up the conclusion section and further added some discussions.

---

## Referee Report (RR1)

Review "The impact of atmospheric dynamics on vertical cloud overlap over the Tibetan Plateau" by J. Li et al

The study uses 4-years of CloudSat-CALIPSO cloud profiles and a reanalysis to explore cloud overlap over the Tibetan Plateau, and its sensitivity to atmospheric instability and wind shear. With this analysis, they propose a parameterization of cloud overlap for GCMs that takes into account this sensitivity.

I appreciate the authors' diligence in taking into account my reservations and their willingness to redo their analysis with the radar-corrected profiles.
There are issues with the study though, mostly with the presentation of the results, and the organization of the paper, as well as some remaining issues with the language usage.

1. As early as the abstract, quantities used to characterize the degree of overlap of cloud layers are introduced but never explained. These are the parameter $\alpha$ and the decorrelation length L. The authors need to explain what these parameters physically mean. This is to say that the cloud overlap characterization and parameterizations need to be explained at the beginning of the paper. The sooner the jargon is introduced and explained the easier it is to follow the paper. The authors have to realize that not a lot of people are familiar with this formalism, and a reminder is necessary. So, the third paragraph of the introduction should be rewritten and include: 1) what is meant by overlap and the three different types, with reference to papers that actually describe parameterizations; 2) explain the formalism introduced by Hogan and Illingworth (2000) and the two quantities that are used to characterize the overlap and 3) the efforts that have been made to characterize the overlap using observations (e.g. Mace and Benson-Troth 2002) and to improve model representation (e.g. the Di Giuseppe and Tompkins 2015 paper, Shonk et al 2010, etc). Then Explain the distinction between continuous and discontinuous cloud layers (I thought that the exact term was contiguous, and non-contiguous) and that there is a consensus on the fact that discontinuous cloud layers are always randomly overlapping. This way you can focus on only contiguous cloud layers later on.

2. The focus of the paper is not very clear: it starts off as an observational study of overlap over the Tibetan Plateau, but navigates through the best way to analyze the data and then moves on to proposing a new parameterization. I think that the interesting point of the study is to test whether existing overlap parameterizations (e.g. the Di Giuseppe and Tompkins 2015 parameterization) are valid over the Tibetan Plateau, demonstrate that it is having difficulty because the relation between cloud overlap and wind shear is not the same as that used in the DGT15 study, and moreover that by also taking into account instability you actually improve the overlap parameterization there. Actually, I think that it is an interesting result that the Shonk et al 2010 scheme is giving fairly decent results too, when it is only latitude dependent. It would have been interesting though if you could demonstrate that your scheme also works in other parts of the

world, in particular over the tropical oceans. In any case these conclusions should be made more prominent, in both the abstract and the conclusions section.

3. It seems to me that the overall method is very much identical to the method used by Di Giuseppe and Tompkins (2015), in particular the choice of horizontal scale, the choice of threshold for the lidar information and the use of the reanalysis to obtain the large scale atmospheric conditions. Therefore most of section 2 could be significantly simplified by summarizing the Di Giuseppe and Tompkins method and choices.

4. It would be great to see the results of the impact of vertical velocities in the paper rather than in supplementary materials. First there are only 7 figures for now, so more could be added, second Figures 3 and 4 could be put together. Mace et al. (2009) found some connection between the occurrence of maximum overlap and strong ascent over the Tropics. Also, according to Naud et al 2008 there is an impact at a continental site in the US, so I am intrigued as to why this is no longer true over the TP. I also wonder what would happen to the total cloud cover if the overlap was parameterized with instability, wind shear and vertical motion: would this make the difference between parameterized and real cloud cover closer to zero? This would be a more convincing test to decide whether vertical velocity has any impact on cloud overlap, other the Tibetan Plateau and elsewhere.

**Detailed comments:**

Abstract:
1. Line 35: you mention an "overlap parameter" but you have not explain what this is. You might want to add a sentence prior to this one explaining that there is such a parameter to characterize the transition from maximum to random overlap with increasing layer separation. "sensitivity" should be "sensitive"
2. Line 37: what is $\alpha$? See above.
3. Line 39: similar comment to above: what is this decorrelation length?
4. Line 42: "above 1 km" is confusing: since these are layer separations, use "greater than 1 km" instead.

Introduction
5. Line 85-90: the phrase in brackets (L85) is incorrect, please explain here what these three assumptions are and how they relate to the "cloud overlap parameterizations" more explicitly. You have two sentences after that explaining what they do, but only explain what maximum overlap is, not the other two.
6. Line 93: isn't the whole point of the overlap parameterization to help make the radiative budget calculation. Here you write "will also", maybe remove "also"?
7. Line 104: remove "other" before "passive measurements", otherwise it sounds as if radar observations are passive and not active measurements.
8. Line 107-108: add "Mace et al. 2009" in your list of references as they also explore overlap using CloudSat-CALIPSO.
9. Line 130: add "Mace and Zhang 2014" for reference to the GEOPROF-LIDAR product.

Section 2:

10. Section 2.3: The first sentence of the section is mentioning an overlap parameter that has still not been defined. So you need to reorder the section such that the equations come first, then the overlap parameter and decorrelation length are introduced and then you can discuss the importance of horizontal scale. In fact this is discussed in section 2.4, so why not wait until then. My preference would be to have most of this material on the formalism of cloud overlap as early as the introduction (see above).

11. Section 2.4: this is a rather long and confusing section, is this necessary when it seems you are in the end using a similar horizontal scale as in Di Giuseppe and Tompkins 2015? Part of the confusion comes from a lack of distinction between the *horizontal* scale, that is the length of the segment of CloudSat orbit you choose to calculate the cloud cover, and your vertical scale as you mention the larger distance here for Figure 2d. As mentioned above, do you need to discuss "discontinuous" layers when you are only interested in continuous layers?

12. Line 293: what does "is resolvable to approximately 2%" mean?

Section 3:
Section 3.1:

13. Line 333-335: I do not understand this sentence, in particular the phrase "cloud-pair related pentad-averaged the degree of conditional instability…"

14. Line 339: do you really mean May *and* September" or instead "May *to* September"?

15. Line 343: "is" should be "are". Here it might be the case that vertical velocities might be large because of extratropical cyclones or other baroclinic instability which could explain maximum overlap. "the increasing of layer distance" should be "the layer separation increases" (check entire text as this phrase is used a few time).

16. Lines 349-352: here it is also quite possible that other large scale forcings might influence the overlap, this should be considered.

17. Line 360: "cloud layer with large distance" should be "cloud layers with large separations". How large? Greater than 2 km, more?

18. Line 375: this is not exactly true, Naud et al (2008) say that vertical velocities in the tropics are not well captured in reanalysis when convection occurs, however they use them in the midlatitudes.

19. Line 375-384: as mentioned previously, the monthly and zonal variation plots are not sufficient proof that vertical velocity is not impacting the overlap. It was found to be the case in the midlatitude winter over land. At least these figures should be included in the manuscript.

20. Line 381: sensitivity to what? "relative" should be "relatively"

Section 3.2:

21. Line 407: the use of "stable" is not clear, do you mean "uniform"? I would write instead that the "relationship display some variability, in particular spatially and seasonally." Or something like that.

22. Line 454: what does "small cloud cover bias" mean?

23. Line 468: "are still difficult" should be "still have difficulties"

24. Line 470: replace "rare" with "scarcity". I do not understand this statement. Why would overlap representation have anything to do with radiosoundings. I think that you refer to the Di Giuseppe and Tompkins (2015) statement about reanalysis being less reliable in places where assimilation of radiosoundings is scarce. This is because in this case, within the fine scale information from the radiosounding missing, the reanalysis is driven mostly by its model (IFS in the case of ECMWF) and the model has a resolution that is too coarse for small separations. Please elaborate. What is the minimum separation in your study, 250 m?

25. Line 475: the sentence "The biases...distinguishable" does not make sense. Please rewrite.

26. Line 476: "close cloud layers": how close, please specify.

27. Line 477-478: replace "are still cause slightly overestimation" with something like "overestimate total cloud cover slightly". This sentence is unclear.

28. Lines 475-484: this whole paragraph is very hard to follow, please try and clarify.

Conclusions:

29. Line 500: please specify "over the Tibetan Plateau" after "data"

30. Line 506: "greater $\alpha$ values": please explain what this means physically.

31. Line 508: again explain what the decorrelation length is physically

32. Line 536: here I am not sure I understand the logic of these last few lines. Surely, cloud trends over the Plateau were obtained with observations and not models? Or do you mean to say that these trends are in fact obtained from GCMs prediction runs? Please specify. Cloud trends from observations have little to do with overlap.

33. Acknowledgments: please specify the locations of the datasets so readers can find them.

**Typos/language issues:**

Abstract:
- Line 34: "overlapped" should be "overlap". Here and every else in the manuscript, the "increasing of layer distance" is incorrect, it should read "increasing layer separation".
- Line 38: "well agreement" is incorrect, replace with "in good agreement". Add "a" before "multiple linear regression method".

Introduction
- Lines 66-67: this sentence is confusing, "increasing" should be "increase", "became" should be "has" and the last statement is unclear, has the "variation" also weakened?
- Line 72: "such as" is not appropriate here, maybe you mean "For example"?
- Section 2.1, line 160: replace "other radar information" with "the radar information"

Section 3:
- Line 327: replace "occurs" with "during"
- Line 340: "instability" should be "unstable"
- Line 348: replace "of" with "between"
- Line 354: replace "to the south part" with "in the southern part"
- Line 355: replace "instability" with "a relatively more unstable"
- Line 356: add "that" before "enhances"

- Line 359: add "the" before "southern part"
- Line 364: replace "contributed" with "attributed"
- Line 366: add "the" before "accelerated"
- Line 370-371: replace "are still difficult to capture" with "still have difficulties to represent"
- Line 371: "those cloud layer" is plural, i.e. "cloud layers"
- Line 400: replace "relative" with "relatively"
- Line 408: remove "shortly"

Conclusions:
- Line 497: replace "and related to" with "and it impact on"
- Line 503: again rewrite "the increasing of layer distance", not correct phrase.
- Line 506: again, "well agreement" should be "in good agreement"
- Line 507: again, add "a" before "multiple"
- Line 511: again, replace "above" with "greater than"

---

## Author Response (AR4)

**Response to Reviewer #1's Comments:**

**Jiming Li et al. (Author)**

**We are very grateful for the Review #1's detailed comments and suggestions, which help us improve this paper significantly. Based on the two Reviewers' comments and suggestions, we reorganized the introduction and added some interpretations in each section in order to make the manuscript more clear. In addition, some superfluous information in each section was deleted.**

**Important revision includes:**

(1) The structure of the manuscript, especially for the Abstract and Introduction sections, was reorganized in order to make the manuscript more clear.

(2) The physically means of technical terms were added in the Abstract section.

(3) The Introduction section also interpreted the main aims of this study.

(4) In each section, we also added some interpretations about the comments from reviewers.

(5) Based on the comments from editor and reviewers, we also corrected the grammars in the revised paper.

**Please see our point-by-point reply to comments. In addition, all revisions were highlighted in revised manuscript by using yellow color.**

**Specific responses:**

(1) As early as the abstract, quantities used to characterize the degree of overlap of cloud layers are introduced but never explained. These are the parameter $\alpha$ and the decorrelation length $L$. The authors need to explain what these parameters physically mean. This is to say that the cloud overlap characterization and parameterizations need to be explained at the beginning of the paper. The sooner the jargon is introduced and explained the easier it is to follow the paper. The authors have to realize that not a lot of

people are familiar with this formalism, and a reminder is necessary. So, the third paragraph of the introduction should be rewritten and include: 1) what is meant by overlap and the three different types, with reference to papers that actually describe parameterizations; 2) explain the formalism introduced by Hogan and Illingworth (2000)and the two quantities that are used to characterize the overlap and 3) the efforts that have been made to characterize the overlap using observations (e.g. Mace and Benson-Troth 2002) and to improve model representation (e.g. the Di Giuseppe and Tompkins2015 paper, Shonk et al 2010,etc). Then explain the distinction between continuous and discontinuous cloud layers (I thought that the exact term was contiguous, and noncontiguous) and that there is a consensus on the fact that discontinuous cloud layers are always randomly overlapping. This way you can focus on only contiguous cloud layers later on.

**Response:** We very thank reviewer for providing detailed comments and suggestions. Based on these suggestions, we reorganized the structure of the Introduction section in order to make the manuscript more clear.

**The second and third paragraphs in the revised manuscript are:**

[revised manuscript text omitted]

(2) The focus of the paper is not very clear: it starts off as an observational study of overlap over the Tibetan Plateau, but navigates through the best way to analyze the data and then moves on to proposing a new parameterization. I think that the interesting point of the study is to test whether existing overlap parameterizations (e.g. the Di Giuseppe and Tompkins 2015 parameterization) are valid over the Tibetan Plateau, demonstrate that it is having difficulty because the relation between cloud overlap and wind shear is not the same as that used in the DGT15 study, and moreover that by also taking into account instability you actually improve the overlap parameterization there. Actually, I think that it is an interesting result that the Shonk et al 2010 scheme is giving fairly decent results too, when it is only latitude dependent. It would have been interesting though if you could demonstrate that your scheme also works in other parts of the world, in particular over the tropical oceans. In any case these conclusions should be made more prominent, in both the abstract and the conclusions section.

**Response:** We very thank reviewer for providing detailed comments and suggestions. Studies have showed that the changes of cloud cover are responsible for the rapid climate warming over the Tibetan Plateau (TP) in the past three decades. It means that the reliable

simulation of cloud cover in the climate models will favor the prediction of climate change over TP. However, our incomplete understanding of the cloud physical processes

125 and the limited cloud observations over the TP make the simulation of total cloud cover in the climate models still unreliable. One of the remaining challenges involves how to reasonably represent the characteristics of the vertical overlapping of cloud layers in these models. Thus, **the main aim of this study is to examine the cloud overlaps over the TP region, and further build an empirical relationship between cloud overlap**

130 **properties and large-scale atmospheric dynamics** by using 4 years (2007–2010) of data from the CloudSat cloud product and collocated ERA-Interim reanalysis product. Recent study has discussed the impact of wind shear on the cloud overlap parameter (Di Giuseppe and Tompkins, 2015). As we know, the TP during summer is usually considered to be an atmospheric heat source or "air pump" due to its higher surface

135 temperature compared with surrounding regions at the same altitude. Additionally, a humid and warm air intrudes from the South Asia monsoon area into the lower atmosphere over the TP to intensify the atmospheric instability of moist convection when combined with the enhanced surface heating. This feature favors the development of convective clouds. It means that the impact of atmospheric instability on the cloud

140 overlap properties should be considered in parameterization. Although previous studies have verified the importance of instability on the cloud overlap properties, its impact wasn't included in the parameterizations of decorrelation length scale $L$. Therefore, the **key focus of this paper** is to develop a new scheme, which considered decorrelation length scale $L$ as a function of the wind shear and atmospheric stability. Our results

145 indicated that new scheme may improve the prediction of cloud cover over TP compared with wind shear-dependent scheme or other schemes. The suggestions from reviewer are very important to us. However, as stated in our paper, current results can't suggest our parameterization was superior in other regions. At present, we are performing another cloud overlap analysis by combining the effects of wind shear, atmospheric stability and

150 vertical velocity in the parameterization. Meantime, the effects of precipitation and cloud system scale also will be considered. Thus, we hope that we may answer the question of reviewer in the further work.

(3) It seems to me that the overall method is very much identical to the method used by Di Giuseppe and Tompkins (2015), in particular the choice of horizontal scale, the choice of threshold for the lidar information and the use of the reanalysis to obtain the large scale atmospheric conditions. Therefore most of section 2 could be significantly simplified by summarizing the Di Giuseppe and Tompkins method and choices.

**Response:** We agreed with reviewer. In the revised manuscript, we simplified the section 2.3 and 2.4 based on the suggestion of reviewer (please see the section 2.3 and 2.4), but some important information was still kept for the readability of manuscript, especially the retrieval method of overlap parameter.

(4) It would be great to see the results of the impact of vertical velocities in the paper rather than in supplementary materials. First there are only 7 figures for now, so more could be added, second Figures 3 and 4 could be put together. Mace et al. (2009) found some connection between the occurrence of maximum overlap and strong ascent over the Tropics. Also, according to Naud et al 2008 there is an impact at a continental site in the US, so I am intrigued as to why this is no longer true over the TP. I also wonder what would happen to the total cloud cover if the overlap was parameterized with instability, wind shear and vertical motion: would this make the difference between parameterized and real cloud cover closer to zero? This would be a more convincing test to decide whether vertical velocity has any impact on cloud overlap, other the Tibetan Plateau and elsewhere.

**Response:** We very thank reviewer for providing detailed comments and suggestions.

[Figure]

Fig.s1.The zonal difference of cloud cover between calculated and observed for different schemes and its variation with layer separations. The Li/Wind-Instability-vertical velocity

180 and Li/Wind-Instability Schemes are from our study.

Indeed, several previous studies have found the impact of vertical velocity on the cloud overlap parameter over Tropics and mid-latitude site. In the revised manuscript, we
185 added the effect of vertical velocity in the Fig 3, Fig4 and Fig5. Based on the suggestion from reviewer, we also parameterized the overlap parameter as a function of instability, wind shear and vertical velocity (see the Figure s1). We agreed with the comments of reviewer. From the Fig.1, we can see that if we combined the impacts of wind shear, instability and vertical velocity on the overlap parameter in the parameterization of
190 decorrelation length scale $L$, the Li/Wind-Instability-vertical velocity produces better cloud cover prediction than Li/Wind-Instability Scheme when cloud layer separations are smaller than 1 km. As the cloud layer separation exceeds 1 km, the biases obviously increase. Compared with Li/Wind-Instability scheme, Li/Wind-Instability-vertical velocity scheme has a relatively lower R-squared values ($R^2$=0.89). Although our results
195 indicated that the vertical velocity at 500hPa has an effect on the cloud overlap parameter, especially for small cloud layer separation, the Li/Wind-Instability-vertical velocity scheme doesn't show better superiority than Li/Wind-Instability Scheme, at least over the TP region. The current study only considered the 500hPa vertical velocity, we are performing another cloud overlap analysis by using the vertical velocity at different
200 levels to determine whether includes the vertical velocity at different levels into the scheme may improve the cloud cover predictions over mid-latitude. **In addition, as stated by reviewer, Naud et al (2008) indicated that vertical velocities are not well captured in the reanalysis when convection occurs, while the convective clouds are very frequent during summer over the Tibetan Plateau. As a result, we only**
205 **parameterized decorrelation length scale $L$ as a function of the wind shear and atmospheric stability in current study.**

**Point-by-point response:**
(5) Line 35: you mention an "overlap parameter" but you have not explain what this is.
210 You might want to add a sentence prior to this one explaining that there is such a parameter to characterize the transition from maximum to random overlap with

increasing layer separation. "sensitivity" should be "sensitive".

**Response:** In the abstract part of the revised manuscript, we added one sentence to interpret the physical meanings of overlap parameter and decorrelation length, respectively. That is, "To do this, the cloud overlap parameter $\alpha$, which is an inverse exponential function of the cloud layer separation $D$ and decorrelation length scale $L$, is calculated and discussed. The parameter $\alpha$ and $L$ are both widely used to characterize the transition from the maximum to random overlap assumption with increasing layer separations".

(6) Line 42: "above 1 km" is confusing: since these are layer separations, use "greater than1 km" instead.

**Response:** We corrected the ambiguous words in the revised manuscript.

(7) Line 85-90: the phrase in brackets (L85) is incorrect, please explain here what these three assumptions are and how they relate to the "cloud overlap parameterizations" more explicitly. You have two sentences after that explaining what they do, but only explain what maximum overlap is, not the other two.

**Response:** In the revised manuscript, we reorganized the introduction section and added some explanations about different cloud overlap assumptions. That is:

"To derive the reasonable total cloud cover between cloud layers, models have to make some assumption about the cloud layers how to overlap in the vertical direction, such as, maximum, random and minimum assumptions. If the cloud covers of two model layers are given by $C_i$ and $C_j$, respectively, total cloud cover between these two layers from maximum assumption is $C_{i,j}^{\max} = \max\{C_i, C_j\}$, while the random and minimum assumptions define the total cloud cover as $C_{i,j}^{ran} = C_i + C_j - C_i \times C_j$ and $C_{i,j}^{\min} = \min\{C_i + C_j, 1\}$, respectively. Thus, the maximum assumption minimizes the total cloud cover, while minimum assumption produces minimally overlap between cloud layers and results in maximum total cloud cover (Weger et al., 1992). The total cloud cover predicted by the random assumption will fall somewhere between maximum and minimum assumption (Geleyn and Hollingsworth, 1979)".

(8) Line 93: isn't the whole point of the overlap parameterization to help make the radiative budget calculation. Here you write "will also", maybe remove "also"?

**Response:** It was removed in the revised manuscript.

(9) Line 104: remove "other" before "passive measurements", otherwise it sounds as if radar observations are passive and not active measurements.

**Response:** Related information was removed in the revised manuscript.

(10) Line 107-108: add "Mace et al. 2009" in your list of references as they also explore overlap using CloudSat-CALIPSO.

**Response:** We added the reference in the revised manuscript.

(11) Line 130: add "Mace and Zhang 2014" for reference to the GEOPROF-LIDAR product.

**Response:** We added the reference in the revised manuscript.

(12) Section 2.3: The first sentence of the section is mentioning an overlap parameter that has still not been defined. So you need to reorder the section such that the equations come first, then the overlap parameter and decorrelation length are introduced and then you can discuss the importance of horizontal scale. In fact this is discussed in section 2.4, so why not wait until then. My preference would be to have most of this material on the formalism of cloud overlap as early as the introduction (see above).

**Response:** We reorganized the sections 2.3 and 2.4, and moved some important information to the second and third paragraphs of the introduction part.

(15) Section 2.4: this is a rather long and confusing section, is this necessary when it seems you are in the end using a similar horizontal scale as in Di Giuseppe and Tompkins 2015? Part of the confusion comes from a lack of distinction between the horizontal scale, that is the length of the segment of CloudSat orbit you choose to calculate the cloud cover, and your vertical scale as you mention the larger distance here for Figure 2d. As mentioned above, do you need to discuss "discontinuous" layers when you are only

interested in continuous layers?

**Response:** Following the comments from two reviewers, we deleted some superfluous information and meanwhile kept some details. In addition, we also added a few explanations about distinction between the horizontal scale of cloud system and spatial scale in order to make this version more readable. The cloud overlap properties over the TP have received little attention. Thus, we still kept a little bit discussion about noncontiguous layers in the revised paper.

(16) Line 293: what does "is resolvable to approximately 2%" mean?

**Response:** Because the along-track resolution of the CPR measurements is about 1.1 km, we used 50 CloudSat profiles as a surrogate of the spatial scale of 50 km. It means that the each cloudy CloudSat profile has a cloud cover about 2% for given spatial scale of 50 km. That is, cloud cover is resolvable to approximately 2%. We added a little bit explanation in the revised manuscript.

(17) Line 333-335: I do not understand this sentence, in particular the phrase "cloud-pair related pentad-averaged the degree of conditional instability…"

**Response:** We replaced this sentence with" Figures 3c and 3e show the monthly variations in pentad-averaged conditional instability of the moisture convection ($\partial\theta_{es}/\partial z$) and the wind shear ($dV/dz$) for the contiguous cloud-pairs over the TP, respectively".

(18) Line 339: do you really mean "May and September" or instead "May to September"?

**Response:** We replaced the "May and September" with "May to September" in the revised manuscript.

(19) Line 343: "is" should be "are". Here it might be the case that vertical velocities might be large because of extratropical cyclones or other baroclinic instability which could explain maximum overlap. "the increasing of layer distance" should be "the layer separation increases" (check entire text as this phrase is used a few time).

**Response:** We added the suggestion from reviewer and corrected the phrase in the revised manuscript.

305 (20) Lines 349-352: here it is also quite possible that other large scale forcings might influence the overlap, this should be considered.

**Response:** Thanks for your comments. We added the suggestion of reviewer in the revised manuscript. Indeed, current investigation only considers the impact of wind shear, instability and vertical velocity on cloud overlap. The effects of other large scale forcings

310 will be considered in further study.

(21) Line 360: "cloud layer with large distance" should be "cloud layers with large separations". How large? Greater than 2 km, more?

**Response:** It was corrected in the revised manuscript and the separations are greater than

315 2km.

(22) Line 375: this is not exactly true, Naud et al (2008) say that vertical velocities in the tropics are not well captured in reanalysis when convection occurs, however they use them in the mid-latitudes.

320 **Response:** We agreed with reviewer. In the revised manuscript, this sentence was corrected as "For example, Naud et al. (2008) indicated that vertical velocities in the tropics are not captured in the reanalysis dataset when convection occurs, thus they only discussed the impact of vertical velocity on the cloud overlap parameter over the mid-latitude and found that vigorous ascent tends to favor maximum overlap."

325

(23) Line 375-384: as mentioned previously, the monthly and zonal variation plots are not sufficient proof that vertical velocity is not impacting the overlap. It was found to be the case in the mid-latitude winter over land. At least these figures should be included in the manuscript.

330 **Response:** We agreed with reviewer. In the revised manuscript, we included the impact of vertical velocities in the text and added further discussion in the section 3.1. (See the revised manuscript). For the related response, please see the reply of question (4) (Line: 174-204).

335 (24) Line 381: sensitivity to what? "relative" should be "relatively"

**Response:** It was corrected in the revised manuscript.

(25) Line 407: the use of "stable" is not clear, do you mean "uniform"? I would write instead that the "relationship display some variability, in particular spatially and

340 seasonally" Or something like that.

**Response:** Based on the suggestion of reviewer, we added the sentence: "relationship display some variability, in particular spatially and seasonally" in the revised manuscript.

(26) Line 454: what does "small cloud cover bias" mean?

345 **Response:** We replaced the sentence with "Compared with random and maximum assumptions, the differences of cloud over caused by other schemes are small and range from -3% to 3%".

(27) Line 468: "are still difficult" should be "still have difficulties"

350 **Response:** It was corrected in the revised manuscript.

(28) Line 470: replace "rare" with "scarcity". I do not understand this statement. Why would overlap representation have anything to do with radio soundings. I think that you refer to the Di Giuseppe and Tompkins (2015) statement about reanalysis being less

355 reliable in places where assimilation of radio soundings is scarce. This is because in this case, within the fine scale information from the radio sounding missing, the reanalysis is driven mostly by its model (IFS in the case of ECMWF) and the model has a resolution that is too coarse for small separations. Please elaborate. What is the minimum separation in your study, 250 m?

360 **Response:** In the revised paper, we deleted the inaccurate presentations. In addition, the minimum separation in my study is 250m.

(29) Line 475: the sentence "The biases…distinguishable" does not make sense. Please rewrite.

365 **Response:** In the revised manuscript, we replaced the sentence with "The differences of

cloud cover caused by different overlap schemes are distinguishable"

(30) Line 476: "close cloud layers": how close, please specify.

**Response:** It was specified in the revised manuscript.

370

(31) Line 477-478: replace "are still cause slightly overestimation" with something like "overestimate total cloud cover slightly". This sentence is unclear.

**Response:** We replaced the "are still cause slightly overestimation" with "overestimate total cloud cover slightly" in the revised manuscript.

375

(32) Lines 475-484: this whole paragraph is very hard to follow, please try and clarify.

**Response:** We reorganized whole paragraph in the revised manuscript.

(33) Line 500: please specify "over the Tibetan Plateau" after "data"

380 **Response:** It was added in the revised manuscript.

(34) Line 506: "greater $\alpha$ values": please explain what this means physically.

**Response:** We added the explanation about overlap parameter $\alpha$ in the revised manuscript.

385

(35) Line 508: again explain what the decorrelation length is physically

**Response:** We added the explanation about decorrelation length in the revised manuscript.

390 (36) Line 536: here I am not sure I understand the logic of these last few lines. Surely, cloud trends over the Plateau were obtained with observations and not models? Or do you mean to say that these trends are in fact obtained from GCMs prediction runs? Please specify. Cloud trends from observations have little to do with overlap.

**Response:** We agreed with reviewer. The last several sentences was corrected as "By
395 using surface observations over 71 stations, some studies verified that annual and seasonal total cloud covers have declined during 1961-2005 (Duan and Wu, 2006; You et

al., 2014). However, whether such variations of total cloud covers are linked with the changes of degree of cloud overlap over the TP are still unclear. Thus, more efforts are needed to reasonably evaluate the impact of cloud overlap on the total cloud cover variations over these sensitive areas of climatic change (e.g., Tibetan Plateau and Arctic)"

(37) Acknowledgments: please specify the locations of the datasets so readers can find them.

**Response:** We added the available links about datasets in the Acknowledgments part.

(38) Line 34: "overlapped" should be "overlap". Here and every else in the manuscript, the "increasing of layer distance" is incorrect, it should read "increasing layer separation".

**Response:** It was corrected in the revised manuscript.

(39) Line 38: "well agreement" is incorrect, replace with "in good agreement". Add "a" before "multiple linear regression method".

**Response:** These errors were corrected in the revised manuscript.

(40) Lines 66-67: this sentence is confusing, "increasing" should be "increase", "became" should be "has" and the last statement is unclear, has the "variation" also weakened?

**Response:** These errors were corrected in the revised manuscript.

(41) Line 72: "such as" is not appropriate here, maybe you mean "For example"?

**Response:** It was corrected in the revised manuscript.

(42) Line 160: replace "other radar information" with "the radar information"

**Response:** We replaced "other radar information" with "the radar information" in the revised manuscript.

(43) Line 327: replace "occurs" with "during"

**Response:** We replaced "occurs" with "during" in the revised manuscript.

(44) Line 340: "instability" should be "unstable"

430 **Response:** We replaced "instability" should be "unstable" in the revised manuscript.

(45) Line 348: replace "of" with "between"
**Response:** We replaced "of" with "between" in the revised manuscript.

435 (46) Line 354: replace "to the south part" with "in the southern part"
**Response:** We replaced "to the south part" with "in the southern part" in the revised manuscript.

(47) Line 355: replace "instability" with "a relatively more unstable"
440 **Response:** We replaced "instability" with "a relatively more unstable" in the revised manuscript.

(48) Line 356: add "that" before "enhances"
**Response:** It was added in the revised manuscript.

445

(49) Line 359: add "the" before "southern part"
**Response:** It was added in the revised manuscript.

(50) Line 364: replace "contributed" with "attributed"
450 **Response:** We replaced "contributed" with "attributed" in the revised manuscript.

(51) Line 366: add "the" before "accelerated"
**Response:** It was added in the revised manuscript.

455 (52) Line 370-371: replace "are still difficult to capture" with "still have difficulties to represent"
**Response:** We replaced "are still difficult to capture" with "still have difficulties to represent" in the revised manuscript.

460 (53) Line 371: "those cloud layer" is plural, i.e. "cloud layers"

**Response:** It was corrected in the revised manuscript.

(54) Line 400: replace "relative" with "relatively"

**Response:** We replaced "relative" with "relatively" in the revised manuscript.

465

(55) Line 408: remove "shortly"

**Response:** It was removed in the revised manuscript.

(56) Line 497: replace "and related to" with "and it impact on"

470 **Response:** We replaced "and related to" with "and it impact on" in the revised manuscript.

(57) Line 503: again rewrite "the increasing of layer distance", not correct phrase.

**Response:** It was corrected in the revised manuscript.

475

(58) Line 506: again, "well agreement" should be "in good agreement"

**Response:** It was corrected in the revised manuscript.

(59) Line 507: again, add "a" before "multiple"

480 **Response:** It was added in the revised manuscript.

(60) Line 511: again, replace "above" with "greater than"

**Response:** We replaced "above" with "greater than" in the revised manuscript.

485

490

**Response to Reviewer #2's Comments:**

**Jiming Li et al. (Author)**

495

**We are very grateful for the Review #2's detailed comments and suggestions, which help us improve this paper significantly. Based on the two Reviewers' comments and suggestions, we reorganized the introduction and added some interpretations in each section in order to make the manuscript more clear. In addition, some**

500 **superfluous information in each section was deleted.**

**Important revision includes:**

(1) The structure of the manuscript, especially for the Abstract and Introduction sections, was reorganized in order to make the manuscript more clear.

505 (2) The physically means of technical terms were added in the Abstract section.

(3) The Introduction section also interpreted the main aims of this study.

(4) In each section, we also added some interpretations about the comments from reviewers.

(5) Based on the comments from editor and reviewers, we also corrected the grammars in

510 the revised paper.

**Please see our point-by-point reply to comments. In addition, all revisions were highlighted in revised manuscript by using yellow color.**

515

**Specific responses:**

**(1) Distance between layers**

It is not entirely clear how cloud populations are separated by distance between layers. If a continuous cloud layer stretches over 6 km in depth, does it contribute to all distances

520 from 0-6km, or only the maximum distance? If all distances, does it therefore contribute to the "1 km distance" multiple times? That is, in a 6-km deep cloud, we can identify 6

pairs of layers that are separated by 1 km. This issue should be addressed when describing the methodology (see also comment about Figure 1, line 213-221).

**Response:** Yes, for the contiguous cloud layer which stretches over 6 km in depth, it contributes to all distances and multiple times for a given distance (e.g., 1 km distance). In the revised manuscript, we added some explanations in the section 2.3:" Layers are analyzed in pairs and no 'double-counting'. If cloud layer pairs have the same separation distance but different altitudes, they will be categorized into the same statistic group". The methodology is same with those of used in previous studies (e.g., Hogan and Illingworth, 2000; Di Giuseppe and Tompkins, 2015).

**(2)** The second issue is that multiple cloud types – or clouds at different heights in the atmosphere – may be grouped together. For instance, a contiguous cloud layer that is 1 km deep can occur anywhere, from stratocumulus, to altostratus, to cirrus. I would expect the shear and stability calculated over 1 km to differ a lot for these different cloud types. The authors should address this, perhaps in a brief discussion section.

**Response:** We very thank for reviewer' comments and suggestions. Indeed, the shear and stability calculated over 1 km should be different for different cloud types. Thus, we added a little bit discussion about uncertainty in the section 4. The added information is **"**In addition, Li et al. (2015) indicated that the overlap properties between different cloud types are also important for the Earth's climate system. Although this study doesn't include the information of cloud type, the dependence of $\alpha$ on meteorological parameters found in our analysis actually demonstrates the effects of cloud types on the $\alpha$ because different combinations of cloud type with the same layer separation possibly occurring  in distinct wind shear and stability conditions**"**.

(3) **Calculation of cloud cover using parameterization schemes**

Section 3.2 is very interesting, but it is sprung upon the reader. The inverse exponential function is not previously introduced. The authors do not explain how the cloud cover is calculated from the different parameterization schemes. Presumably, the decorrelation length scale L is calculated from the dV/dz and dtheta/dz derived from ERA-Interim data interpolated to the CloudSat track. Subsequently, alpha can be calculated for each

separation using equation 6. But how does this lead to a calculation of cloud cover (which is compared in Figure 6, according to line 450)? One additional paragraph in this section (prior to presentation of Figure 6) describing the cloud cover calculation is required.

**Response:** We agreed with reviewer. Based on the comments and suggestions from two reviewers, we reorganized the third paragraph of the introduction to make the definitions of overlap parameter $\alpha$ and decorrelation length scale $L$ more clear.

**We added the sentence** "Finally, Hogan and Illingworth (2000) fitted the reduction in $\alpha$ with layer separation $D$ as an inverse exponential function of the decorrelation length scale $L$: $\alpha = e^{-D/L}$. Thus, $\alpha$ and $L$ are both used to characterize the transition from the maximum to random overlap assumption with increasing layer separations." in the third paragraph of introduction part in the revised manuscript.

**In addition, we also added one additional paragraph in this section 4 in order to describe the cloud cover calculation.** "After deriving the regression formula of decorrelation length scale $L$, we re-apply it to all contiguous cloud samples and retrieve the $L$ and corresponding $\alpha$ based on the formula: $\alpha = e^{-D/L}$ and dynamical conditions. Finally, retrieved overlap parameter $\alpha$ is used to calculate the total cloud cover between any two cloud layers by using the Equ. (1) and the definitions of random and maximum overlap assumptions".

**Point-by-point response:**

(4) Line 27: One sentence in the abstract on the importance of understanding cloud overlap in the Tibetan Plateau would draw in a broader audience.

**Response:** We reorganized the abstract part in the revised paper. The beginning of the abstract is "Studies have showed that the changes of cloud cover are responsible for the rapid climate warming over the Tibetan Plateau (TP) in the past three decades. To simulate the total cloud cover, atmospheric models have to reasonably represent the characteristics of vertical overlap between cloud layers."

(5) Line 32: "Unique" suggests that the authors have compared the TP to all other regions. Perhaps remove this sentence.

**Response:** It was removed in the revised manuscript.

585      (6) Line 68: "Kang et al summarized" – based on what? Observations? Models?

**Response:** It was corrected in the revised manuscript. See the Introduction section.

(7) Line 82 & 85: Please specify what type of models you are mostly concerned with, e.g. "horizontal grid length greater than 10 km". So that it is clear to the reader that only those

590      models rely on some overlap parameterization.

**Response:** Our study focused on the climate change, thus climate models is the first choice. In the revised manuscript, we specified it.

(8) Line 106: Are there any radar sites in the TP region at all?

595      **Response:** There is no long-term Radar observation over the TP region.

(9) Line 113: Remove "fortunately".

**Response:** It was removed in the revised manuscript.

600      (10) Line 124: Break paragraph at "However, the related question…" and merge the remainder with the following paragraph. That sentence is a clear purpose of the paper.

**Response:** We agreed with reviewer. It was corrected in the revised manuscript.

(11) Line 148: Remove "can".

605      **Response:** It was removed in the revised manuscript.

(12) Line 160: At this point, it is important to clarify that the radar does not distinguish between cloud and precipitation. Then the text at 320-332 will not be such a surprise.

**Response:** We agreed with reviewer. We added the explanation: "It is noting that the

610      2B-GEOPROF-LIDAR dataset does not distinguish cloud and precipitation, therefore any bias in our results caused by precipitation can't be removed in current analysis." in the revised manuscript.

(13) Line 165: Add here the lines 196-199 regarding noise in the observations and surface

615  contamination.

**Response:** It was added in the revised manuscript. See section 2.1.

(14) Line 213-221 & Figure 1: This is rather difficult to follow. It would be useful to have one or two additional panels that illustrate the cloud cover and overlap parameter for the

620  particular scene, perhaps for the different length scales considered. For instance, it would be great to have an example of a continuous cloud that is more than 4 km deep, so that the reader can see how it may have "less than random" overlap.

**Response:** We agreed with reviewer. In the revised manuscript, we reorganized the sections 2.3 and 2.4. In addition, we also added two additional subplots in the figure1 to

625  illustrate the cloud cover and overlap parameter for the given cloud scene. Please see the sections 2.3 and 2.4.

(15) Line 217: What does "cloudy" mean in this context?

**Response:** The "cloudy" means that the atmospheric layers are classified as cloud layers.

630  In the revised manuscript, we replaced "cloudy" with "cloud layers".

(16) Line 233: Please change all references to "discontinuous" to "non-continuous", or adjust your figures (be consistent).

**Response:** In the revised manuscript, we replaced "discontinuous" with "noncontiguous

635  ".

(17) Line 248: "correlation" – no correlation has been shown or calculated.

**Response:** It was corrected in the revised manuscript.

640  (18) Line 253-254: Remove "provided … error. Simply these authors"

**Response:** It was removed in the revised manuscript.

(19) Line 260-261: "should account for the typical cloud system scales" add "in their parameterization schemes" (presumably).

645  **Response:** It was added in the revised manuscript.

(20) Line 281: "the number of available samples" – What is a sample? Is it a 50-km stretch in a CloudSat orbit?

**Response:** In the revised manuscript, we replaced "the number of available samples" with "the number of available cloud pair samples". Thus, the number is just for the cloud layer-pair, not for the 50-km segment in a CloudSat orbit.

(21) Line 281-294: Figure 2d is rather difficult to interpret. It is likely that these values make more sense when they are presented in a table.

**Response:** In revised manuscript, we added some explanations about sample number in order to make the sentence more clear. That is, "Fig.2d shows the variations of sample number and the cumulative percentage with cloud layer separation for both noncontiguous and contiguous clouds at a given spatial scale of 50 km. It shows that the cumulative proportion of cloud sample significantly increases with increasing layer separation. For the contiguous cloud, the cumulative percentage accounts for 90% of all samples when layer separation is smaller than 4 km".

(22) Line 281-294: The description of cloud cover (versus cloud fraction) would have been helpful sooner, probably around line 213-221 (possibly in combination with an illustration in Figure 1).

**Response:** We agreed with reviewer. In the revised manuscript, we moved the description of cloud cover and cloud fraction to the beginning of section 2.3.

(23) Line 301: "thicker than other seasons" – is there a simply explanation for this, e.g. a greater tropopause height?

**Response:** We added a simply explanation about this in the revised manuscript. That is, "frequent strong convective motions during summer season favor deep cloud systems".

(24) Line 321: "small horizontal scale of cumulus" – The authors should also comment on the fact that these are poorly observed by CloudSat alone, so require the lidar to be available (not extinguished by cloud aloft). How does that affect the statistics?

**Response:** We agreed with reviewer. Indeed, although the 2B-GEOPROF-lidar dataset includes the lidar information, cumulus with small scale and cloud systems closed the surface are still missed in our study. It means that our statistical results will be slightly underestimated, but this bias can be partly offset by precipitation effect, especially during the summer season. Thus, it isn't the main source of uncertainty in our analysis.

(25) Line 395: Regarding Figure 5, please mention which scale is used for the segments, presumably 50 km?

**Response:** It was added in the revised manuscript.

(26) Line 400: "seems relatively weaker" – this is difficult to quantify when the two parameters have different units.

**Response:** The sentence was removed in the revised manuscript.

(27) Line 411: "As we know" – actually, this is completely new to the reader! (remove).

**Response:** It was removed in the revised manuscript.

(28) Line 481: "this new scheme" – please refer to the name of the scheme.

**Response:** It was corrected in the revised manuscript.

(29) Figure 1: Apart from the comment above (213-221), mention in the caption that observations near the surface have been removed.

**Response:** We already added the sentence in the caption of Figure 1.

(30) Figure 2: What is the uncertainty on alpha? Although the authors provide the sample number in panel d, there could still be a lot of variation in alpha. The authors should provide some measure of uncertainty, e.g. standard deviation or interquartile range.

**Response:** We agreed with reviewer. In the revised manuscript, we added the error bars in the Figure 2 and revised the figure caption.

(31) Figure 5: Again, what is the uncertainty in alpha? The sample number will be

smaller due to the compositing on dV/dz and dtheta/dz. The reference to "50% continuous" is confusing in the legend and should be placed in the caption.

710 **Response:** We agreed with reviewer. In the revised manuscript, we added the error bars in the Figure 5 and reorganized the legends. In addition, we also added the subplots about vertical velocity in the Fig.5. Please see the revised manuscript.

[revised manuscript text omitted]

endpoint of the 99% confidence interval, respectively. ; (c) The probability distribution functions (PDFs) of the along-track horizontal scales of cloud system at different height over TP region; (d) The variations of cloud sample number and the cumulative percentages with cloud layer separations for both noncontiguous and contiguous clouds at a given spatial scale of 50km. The cumulative percentages represent the proportions of cloud sample below corresponding layer separation to all samples.

**Figure 3.** The monthly variations of the pentad-averaged (a) cloud overlap parameter, $\alpha$, (c) conditional instability to moist convection, $\partial\theta_{es}/\partial z$ (K km$^{-1}$), (e) wind shear, $dV/dz$ (m s$^{-1}$ km$^{-1}$), (g) and vertical velocity $\omega$ (hPa/day) at 500 hPa, $\omega$ for the contiguous cloud layers over the TP ; The monthly variations of the pentad-averaged (b) $\alpha$, (d) $\partial\theta_{es}/\partial z$, (f) $dV/dz$ and (h) $\omega$ for the contiguous clouds for the layer separation of 2 km (red) and 3km (black).

**Figure 4.** The zonal variations of the (a) $\alpha$, (c) $\partial\theta_{es}/\partial z$ (K km$^{-1}$), (e) $dV/dz$ (m s$^{-1}$ km$^{-1}$), and (g) $\omega$ (hPa/day) for the contiguous cloud layers over the TP. ; The zonal variations of the (b) $\alpha$, (d) $\partial\theta_{es}/\partial z$, (f) $dV/dz$ and (h) $\omega$ for the contiguous cloud layers for the layer separation of 2 km (red) and 3km (black).

**Figure 5.** The sensitivities of median overlap parameter $\alpha$ to the (a) wind shear, (b) instability and (c) vertical velocity at 500 hPa at a given upper limit of cloud cover (50%) and spatial scale (50 km) for the contiguous cloud layers. The horizontal bars correspond to means±3 standard errors, which represent the upper endpoint and lower endpoint of the 99% confidence interval, respectively. The error bars correspond to ±3 standard error.

**Figure 6.** The monthly differences in total cloud cover (unitless) between calculation and observation for different schemes (see the Table 1) and its dependence on the layer separation.

**Figure 7.** The zonal differences in cloud cover (unitless) between calculation and observation for different schemes (see the Table 1) and its dependence on the layer separation.

1575

[Figure]

**Figure 1.** (a) CloudSat overpass tracks (blue line: daytime; red line: nighttime) over the Tibetan Plateau (27°N-39°N; 78°E-103°E); (b) A sample of CloudSat

1580    2B-GEOPROF-LIDAR cloud mask product along the ground track of 200km (white color: cloud fraction>99%; light blue: 0<cloud fraction<99%; deep blue: clear sky; orange color: surface).==(c) The observed and calculated segment-average cloud cover profiles based on maximum and random assumptions for different spatial scales and given cloud mask sample in Fig. 1b. (d) The corresponding cloud overlap parameters of==

1585    ==contiguous cloud layers for 25, 50, 100 and 200 km spatial scales, respectively. Note that the observations below 1 km over the TP surface have been removed.==

1590

[Figure]

1595

**Figure 2.** The dependence of $\alpha$ on the layer separation and its sensitivity to the spatial scale for (a) noncontiguous and (b) contiguous cloud pairs; The horizontal bars correspond to means $\pm 3$ standard error, which represent the upper endpoint and lower endpoint of the 99% confidence interval, respectively. the error bars correspond to $\pm 3$

1600

standard error; (c) The probability distribution functions (PDFs) of the along-track horizontal scales of cloud system at different height over TP region; (d) The variations of cloud sample number and the cumulative percentages with cloud layer separations for both noncontiguous and contiguous clouds at a given spatial scale of 50km. The cumulative percentages represent the proportions of cloud sample below corresponding

1605

layer separation to all samples.

[Figure]

**Figure 3.** The monthly variations of the pentad-averaged (a) cloud overlap parameter, $\alpha$, (c) conditional instability to moist convection, $\partial\theta_{es}/\partial z$ (K km$^{-1}$) (e) wind shear, $dV/dz$ (m s$^{-1}$ km$^{-1}$), (g) and vertical velocity $\omega$ (hPa/day) at 500 hPa, for the contiguous cloud layers over the TP ; The monthly variations of the pentad-averaged (b) $\alpha$, (d) $\partial\theta_{es}/\partial z$, (f) $dV/dz$ and (h) $\omega$ for the contiguous clouds for the layer separation of 2 km (red) and 3km (black). ~~The monthly variations of the pentad averaged (a) cloud overlap parameter, $\alpha$, (c) conditional instability to moist convection, $\partial\theta_{es}/\partial z$, (e) wind shear, $dV/dz$, (g) and vertical velocity at 500 hPa, $\omega$ for the contiguous cloud layers over the TP ; The monthly variations of the pentad averaged (b) $\alpha$, (d) $\partial\theta_{es}/\partial z$, (f) $dV/dz$ and (h) $\omega$ for the contiguous clouds for the layer separation of 2 km (red) and 3km (black).~~


[Figure]

**Figure 4.** The zonal variations of (a) $\alpha$, (c) $\partial\theta_{es}/\partial z$ (K km$^{-1}$), (e) $dV/dz$ (m s$^{-1}$ km$^{-1}$), and (g) $\omega$ (hPa/day) for the contiguous cloud layers over the TP. The zonal variations of the (b) $\alpha$, (d) $\partial\theta_{es}/\partial z$, (f) $dV/dz$ and (h) $\omega$ for the contiguous cloud layers for the layer separation of 2 km (red) and 3km (black).

**Figure 4.** The zonal variations of the (a) $\alpha$, (e) $\partial\theta_{es}/\partial z$, (e) $dV/dz$, and (g) $\omega$ for the contiguous cloud layers over the TP ; The zonal variations of the (b) $\alpha$, (d) $\partial\theta_{es}/\partial z$, (f) $dV/dz$ and (h) $\omega$ for the contiguous cloud layers for the layer separation of 2 km (red) and 3km (black).

[Figure]

1640 **Figure 5.**The sensitivities of median overlap parameter $\alpha$ to the (a) wind shear, (b) instability and (c) vertical velocity at 500 hPa at a given upper limit of cloud cover (50%) and spatial scale (50 km) for the contiguous cloud layers. The horizontal bars correspond to means±3 standard error, which represent the upper endpoint and lower endpoint of the 99% confidence interval, respectively.The error bars correspond to ±3

1645 standard error.

[Figure]

1650 Figure 6. The monthly differences in cloud cover (unitless) between calculation and observation for different schemes (see the Table 1) and its dependence on the layer separation.

[Figure]

1655

Figure 7. The zonal differences in cloud cover (unitless) between calculation and

observation for different schemes (see the Table 1) and its dependence on the layer separation.